# Memory recall involves a transient break in excitatory-inhibitory balance

**Renée S Koolschijn[1]\*[†], Anna Shpektor[1][†], William T Clarke[1], I Betina Ip[1], David Dupret[2], Uzay E Emir[1,3], Helen C Barron[1,2]\***

[1]Wellcome Centre for Integrative Neuroimaging, University of Oxford, FMRIB, John Radcliffe Hospital, Oxford, United Kingdom; [2]Medical Research Council Brain Network Dynamics Unit, University of Oxford, Oxford, United Kingdom; [3]School of Health Sciences, Purdue University, West Lafayette, United States

**Abstract** The brain has a remarkable capacity to acquire and store memories that can later be selectively recalled. These processes are supported by the hippocampus which is thought to index memory recall by reinstating information stored across distributed neocortical circuits. However, the mechanism that supports this interaction remains unclear. Here, in humans, we show that recall of a visual cue from a paired associate is accompanied by a transient increase in the ratio between glutamate and GABA in visual cortex. Moreover, these excitatory-inhibitory fluctuations are predicted by activity in the hippocampus. These data suggest the hippocampus gates memory recall by indexing information stored across neocortical circuits using a disinhibitory mechanism.

**\*For correspondence:**
renee.koolschijn@keble.ox.ac.uk (RSK);
helen.barron@merton.ox.ac.uk (HCB)

[†]These authors contributed equally to this work

**Competing interest:** The authors declare that no competing interests exist.

## Introduction

Memories are thought to be stored across sparse and distributed neuronal ensembles in the brain (*Buzsáki, 2010*; *Josselyn and Tonegawa, 2020*). During memory recall, activity across these neuronal ensembles is selectively reinstated to recover enduring representations of the past. This reinstatement is thought to be mediated by the hippocampus, a brain region important for learning and memory (*Squire, 1992*). Anatomically, the hippocampus sits at the apex of a cortical sensory processing hierarchy (*Felleman and Essen, 1991*) where inputs received by sensory cortices reach the hippocampus via the entorhinal cortex and other relay regions, which in turn make widespread cortico-cortical connections that project the hippocampal output back to neocortex (*Witter, 1993*; *Witter et al., 1989*). This reciprocal anatomical connectivity equips the hippocampus with the necessary architecture to coordinate activity in neocortex. The hippocampus may therefore be considered to provide a 'memory index', or summary sketch, for information stored across distributed cortical circuits (*Goode et al., 2020*; *Teyler and DiScenna, 1985*; *Teyler and Rudy, 2007*). Consistent with this view, during memory recall, hippocampal reinstatement predicts subsequent neocortical reinstatement (*Pacheco Estefan et al., 2019*; *Tanaka et al., 2014*).

However, the mechanism that allows the hippocampus to coordinate reinstatement across distributed neocortical circuits remains unclear. One possibility is that the hippocampus shapes computations performed by neocortical circuits by modulating the dynamic interplay between excitation and inhibition (EI). At the cellular level, tight coupling between neocortical EI can be observed during both sensory stimulation and spontaneous neural activity (*Haider et al., 2006*; *McCormick et al., 2004*; *Okun and Lampl, 2008*; *Wehr and Zador, 2003*). This phenomenon has led to the physiological concept of EI balance, where, following changes in excitability, synaptic strength, current, or overall network activity returns to a stable set point via negative feedback (*Field et al., 2020*). Evidence in humans, animal models, and theoretical models together suggests that EI balance is maintained to hold memories in a silent and dormant state (*Barron et al., 2016*; *Froemke et al., 2007*; *Vallentin*

**eLife digest** Memories are stored by distributed groups of neurons in the brain, with individual neurons contributing to multiple memories. In a part of the brain called the neocortex, memories are held in a silent state through a balance between excitatory and inhibitory activity. This is to prevent them from being disrupted by incoming information. When a memory is recalled, an area of the brain called the hippocampus is thought to instruct the neocortex to activate the appropriate neuronal network. But how the hippocampus and neocortex coordinate their activity to switch memories 'on' and 'off' is unclear.

The answer may lie in the fact that neurons in the neocortex consist of two broad types: excitatory and inhibitory. Excitatory neurons increase the activity of other neurons. They do this by releasing a chemical called glutamate. Inhibitory neurons reduce the activity of other neurons, by releasing a chemical called GABA. Koolschijn, Shpektor et al. hypothesized that the hippocampus activates memories by changing the balance of excitatory and inhibitory activity in neocortex.

To test this idea, Koolschijn, Shpektor et al. invited healthy volunteers to explore a virtual reality environment. The volunteers learned that specific sounds in the environment predicted the appearance of particular visual patterns. The next day, the volunteers returned to the environment and viewed these patterns again. After each pattern, they were invited to open a virtual box. Volunteers learned that some patterns led to money in the virtual box, while other patterns did not.

Finally, on day three, the volunteers listened to the sounds from day one again, this time while lying in a brain scanner. The volunteers' task was to infer whether each of the sounds would lead to money. Given that the sounds were never directly paired with the content of the virtual box, the volunteers had to solve the task by recalling the associated visual patterns. As they did so, the brain scanner measured their overall brain activity. It also assessed the relative levels of excitatory and inhibitory activity in visual areas of the neocortex, by measuring glutamate and GABA.

The results revealed that as the volunteers recalled the visual cues, activity in both the hippocampus and the visual neocortex increased. Moreover, the ratio of glutamate to GABA in visual neocortex also increased which was predicted by activity in the hippocampus. This suggests that the hippocampus reactivates memories stored in neocortex by temporarily increasing excitatory activity to release memories from inhibitory control.

Disturbances in the balance of excitation and inhibition occur in various neuropsychiatric disorders, including schizophrenia, autism, epilepsy and Tourette's syndrome. Damage to the hippocampus is known to cause amnesia. The current findings suggest that memories may become inaccessible – or may be activated inappropriately – when the interaction between the hippocampus and neocortex goes awry. Future studies could test this possibility in clinical populations.

*et al., 2016*; *Vogels et al., 2011*), thus protecting memories from interference caused by new learning (*Koolschijn et al., 2019*; *Kuchibhotla et al., 2017*). During recall, however, EI balance must be transiently disturbed if memories are to be released from inhibitory control.

Here, we predict that memory recall involves a transient break in EI balance, opening a window to release memories from the blanket of inhibition before network stability is re-established. Moreover, we predict that this transient break in neocortical EI balance is mediated by activity in the hippocampus. To test these predictions, here, we implemented a new imaging sequence in humans that combines functional magnetic resonance imaging (fMRI) with functional magnetic resonance spectroscopy (fMRS) (*Ip et al., 2019*; *Ip et al., 2017*). This sequence provides an opportunity to monitor activity in the hippocampus with fMRI while simultaneously measuring time-resolved fluctuations in neocortical glutamate and GABA using fMRS.

MRS provides a unique tool to quantify the concentration of different neural metabolites (*De Graaf, 2019*; *Mangia et al., 2012*), including glutamate and GABA, the principle excitatory and inhibitory neurotransmitters in the brain. MRS cannot dissociate between neurotransmitter and metabolic pools of glutamate and GABA (*Bak et al., 2006*; *Magistretti and Allaman, 2015*). However, meaningful interpretation of MRS nevertheless derives from a major body of work showing an approximately 1:1 relationship between the rate of glutamine-glutamate cycling, which is necessary for glutamate and GABA synthesis, and neuronal oxidative glucose consumption, which indirectly

supports neurotransmitter release among other processes (*Rothman et al., 2003*; *Shen et al., 1999*; *Sibson et al., 1998*). Therefore, while measures of EI balance vary in both definition and granularity, MRS can provide a non-invasive marker for physiologically relevant EI at a coarse spatiotemporal scale. Correspondingly, MRS-derived glutamate and GABA reported during learning and memory paradigms show remarkable consistency with findings reported at the physiological level in animals (*Barron et al., 2016*; *Castro-Alamancos et al., 1995*; *Floyer-Lea et al., 2006*; *Froemke et al., 2007*; *Kolasinski et al., 2019*; *Lunghi et al., 2015*; *Trepel and Racine, 2000*; *Vallentin et al., 2016*).

Using the combined fMRI-fMRS sequence, here, we implemented a task designed to engage hippocampal-dependent recall of a visual cue. During memory recall, we report a transient increase in the ratio between MRS-derived glutamate and GABA in neocortex which is selectively predicted by the blood oxygen level-dependent (BOLD) signal in the hippocampus. These findings suggest the hippocampus coordinates memory recall by transiently perturbing neocortical EI balance to release memories stored across distributed neural circuits.

## Results

### Task design and behaviour

To investigate the neuronal mechanisms that support memory recall, we designed a three-stage inference task. This task has previously been shown to involve associative memory recall in humans (*Barron et al., 2020*; *Koster et al., 2018*) and mice (*Barron et al., 2020*). Unlike some forms of associative recall, previous lesion and optogenetic studies in rodents demonstrate that associative recall required for inference is a hippocampal-dependent process (*Barron et al., 2020*; *Bunsey and Eichenbaum, 1996*; *DeVito et al., 2010*). Thus, the inference task provides an opportunity to investigate whether activity in the hippocampus mediates dynamic changes in neocortical EI during memory recall.

The inference task was performed in virtual reality (VR) (*Figure 1A*), an immersive and highly controlled 3D environment that has the potential to benefit from cross-species comparisons in the future (*Barron et al., 2020*). The inference task was performed across 3 days and included three stages (*Figure 1B*). In the first stage of the task, participants learned up to 80 auditory-visual associations ('associative learning', day 1; *Figure 1B*, *Figure 1—figure supplement 1*). In the second stage, which occurred approximately 24 hr later, each visual cue was paired with either a rewarding (set 1, monetary reward) or neutral outcome (set 2, woodchip) delivered to a wooden box in the corner of the VR environment ('conditioning', day 2; *Figure 1A–B*, *Figure 1—figure supplement 1*). Auditory cues were never paired with an outcome, providing an opportunity to assess evidence for an inferred relationship between these indirectly related stimuli.

Accordingly, in the third stage of the task, we presented auditory cues in isolation, without visual cues or outcomes, and we measured evidence for inference from the auditory cues to the appropriate outcome ('inference test', day 3; *Figure 1B*). Participants performed the inference test during an MRI scan (*Figure 1C–D*, *Figure 1—video 1*). On each trial of the inference test, participants were presented with an auditory cue, before being asked if they would like to look in the wooden box ('yes' or 'no') where they had previously found the outcomes during the conditioning stage. Participants' responses depended upon whether they *inferred* the indirectly associated outcome to be rewarding or neutral. On trials where the auditory cue was associated with a visual cue paired with a rewarding outcome (set 1 cues), participants were expected to select 'yes' if they inferred the relevant outcome (*Figure 1E*). On trials where the auditory cue was instead associated with a visual cue paired with a neutral outcome (set 2 cues), participants were expected to select 'no' if they inferred the relevant outcome (*Figure 1E*). We thus categorised trials during the inference test as 'correctly inferred' if participants selected 'yes' when the auditory cue was indirectly associated with rewarding outcome or 'no' when the auditory cue was indirectly associated with a neutral outcome.

Previous studies using this task show that in trials where participants infer the correct outcome, the associated visual cue that links the auditory cue and outcome is reinstated in the hippocampus and visual cortex (*Barron et al., 2020*). Consistent with these previous findings, here, we show that participants make the correct inference only if they can later recall the relevant auditory-visual association during a surprise post-scan associative test (*Figure 1C*; *Figure 2A, C*). Indeed, performance on the post-scan associative test that assessed memory for auditory-visual associations learned on day 1 predicted performance on the inference test (*Figure 2*). The inference task thus provides a

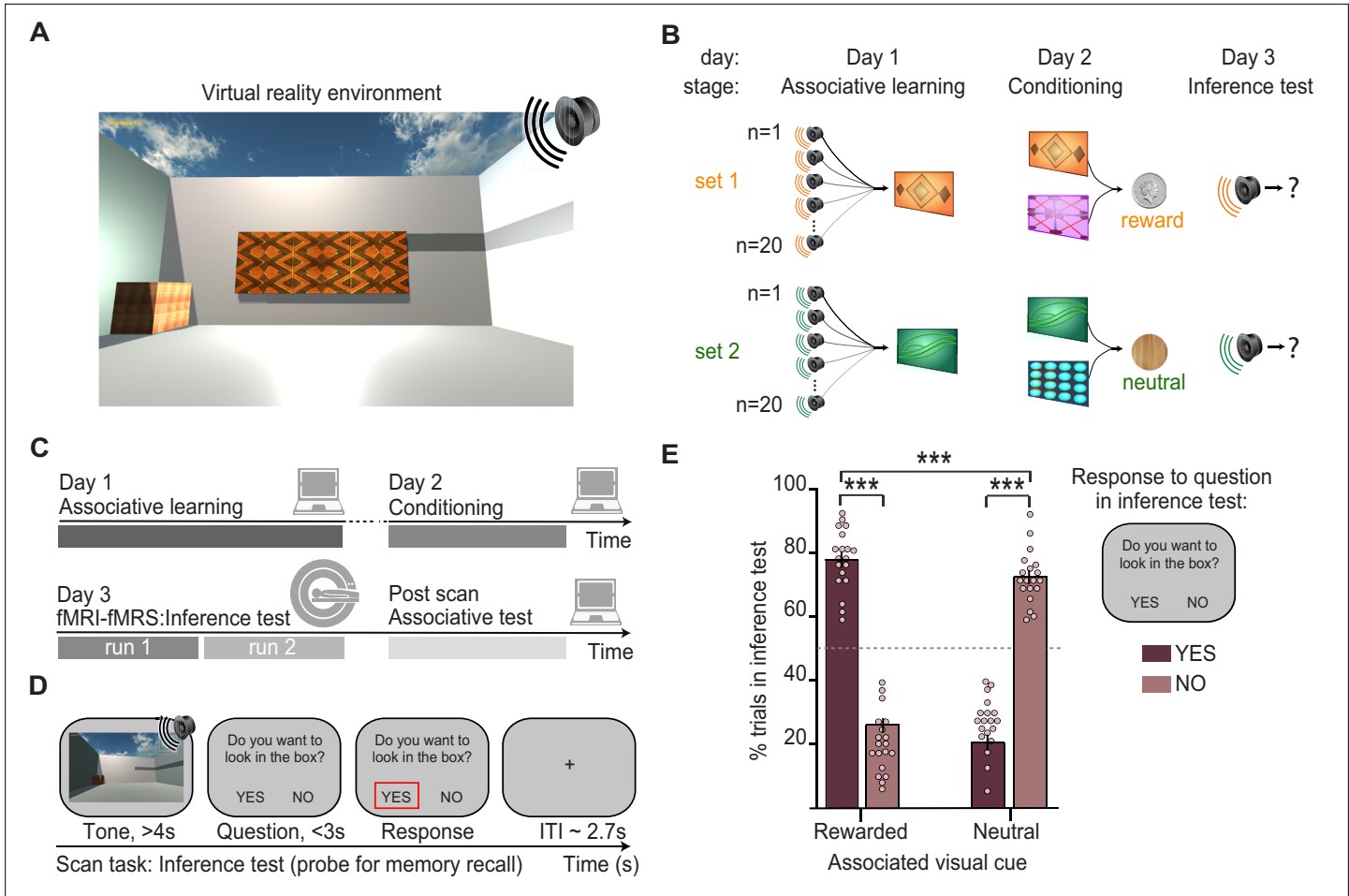

**Figure 1.** Inference task design and behavioural performance. **(A)** The inference task was performed within a virtual-reality environment. **(B)** Three-stage inference task designed to investigate hippocampal-dependent associative memory recall. First, participants learned to associate auditory cues with visual cues ('associative learning' stage, day 1), where four different visual cues were each associated with 20 auditory cues. Second, participants learned to associate visual cues with an outcome delivered to a wooden box in the corner of the virtual-reality environment ('conditioning' stage, day 2). Two visual cues predicted a rewarding outcome (set 1, monetary coin) while the other two predicted a neutral outcome (set 2, woodchip). Third, the auditory cues were played in isolation and we assessed participants' ability to infer the relevant outcome by recalling the intermediary visual cue ('inference test', day 3). **(C)** Schematic: training and testing protocol. The inference test was performed inside the 7T MRI scanner. After exiting the scanner, participants were given a surprise post-scan associative test to directly assess participants' memory for auditory-visual associations learned on day 1. **(D)** Example inference test trial performed inside the scanner. For each auditory cue, participants were required to infer the indirectly associated outcome by recalling the relevant auditory-visual association. To indicate whether participants inferred the outcome to be rewarding or neutral, on each trial participants pressed 'yes' or 'no' in response to a question asking, 'Would you like to look in the box?', referring to the box where the outcome cues were delivered during conditioning. **(E)** Behaviour during the inference test revealed a significant interaction between the response to auditory cues in set 1 and 2 ("rewarded" and "neutral") and whether or not the participants indicated that they wanted to look in the box ("yes" and "no") (two-way ANOVA, $F_{1,72} = 630.99$, p < 0.001). Tukey's post hoc test showed participants pressed "yes" more often for auditory cues in set 1 (p < 0.001), and "no" more often for auditory cues in set 2 (p < 0.001). This inferential behaviour was observed despite participants never experiencing the outcomes in response to the auditory cues. For the purpose of the analyses reported in Figs. 2-5, trials where participants pressed "yes" for auditory cues in set 1, or "no" for auditory cues in set 2 were categorised as "correctly inferred" trials. *** indicates p<0.001.

The online version of this article includes the following video, source data, and figure supplement(s) for figure 1:

**Source data 1.** Percentages of inference test trials in set 1 and 2 ("rewarded" and "neutral") split according to whether participants wanted to look in the box ("yes" and "no").

**Figure supplement 1.** Behavioural training and performance.

**Figure 1—video 1.** Example video used for trials on the inference test during the magnetic resonance imaging (MRI) scan.

https://elifesciences.org/articles/70071/figures#fig1video1

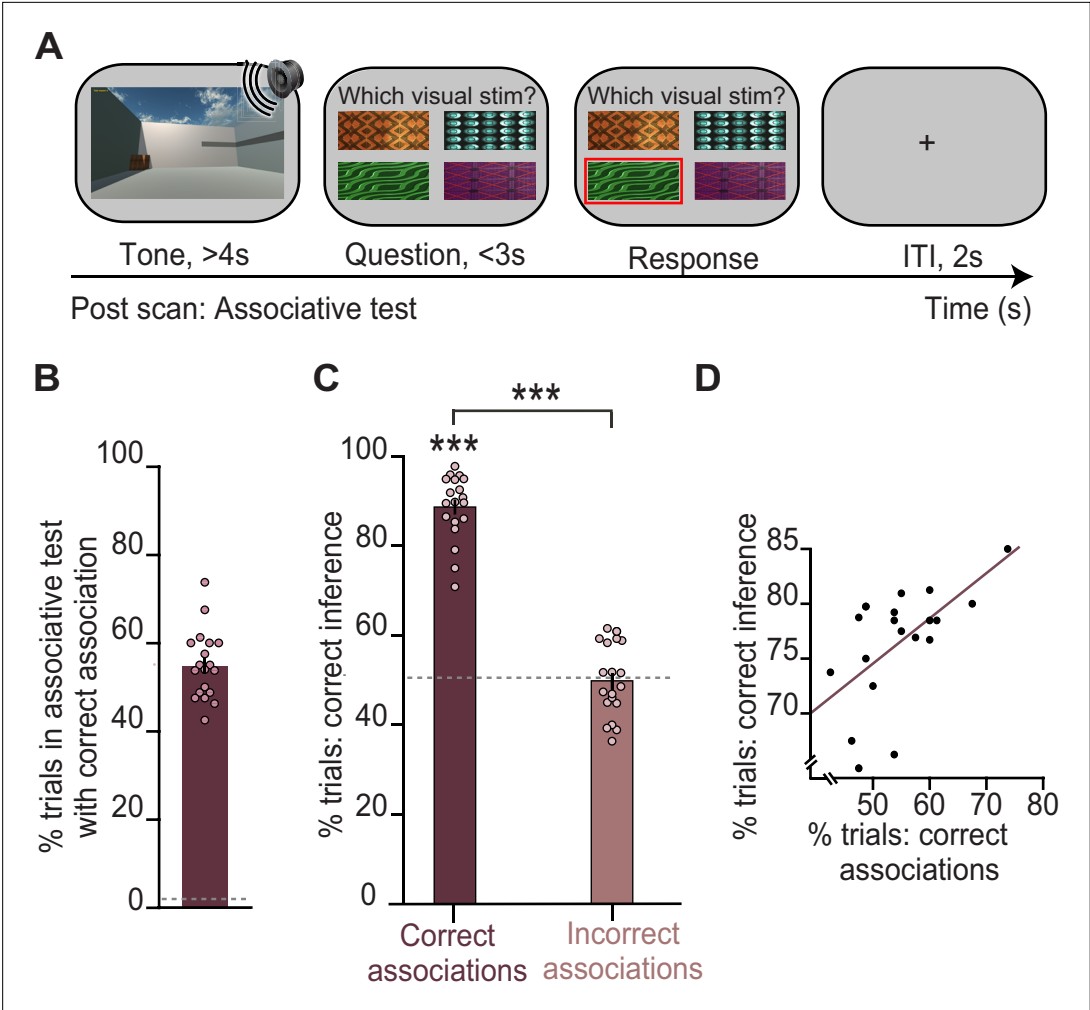

**Figure 2.** Behavioural performance in the inference test is predicted by performance on an associative test. (**A**) Example trial from the surprise post-scan associative test which directly tested participants' memory for auditory-visual associations learned on day 1. (**B**) During the post-scan associative test, participants remembered 55 % of the auditory-visual associations (54.8% ± 1.78%; mean ± SEM), significantly above chance as indicated by the dotted line ($t_{18}$ = 29.96, p < 0.001). (**C**) In response to auditory cues during the inference test (*Figure 1D*), participants successfully inferred the appropriate outcome (*Figure 1E*) on trials where they could later recall the correct auditory-visual association in the post-scan associative test ('correct association': $t_{18}$ = 22.91, p < 0.001; 'incorrect association': $t_{18}$ = 0.09, p = 0.925; 'correct association'–'incorrect association': $t_{18}$ = 16.21, p < 0.001; dotted line indicates chance). (**D**) Across participants, behavioural performance on the inference test was predicted by behavioural performance on the post-scan associative test ($r_{17}$ = 0.57, p = 0.010). Notably, there was no significant effect of sex on behavioural performance (*Supplementary file 1*). *** indicates p < 0.001.

The online version of this article includes the following source data for figure 2:

**Source data 1.** Percentages of remembered auditory-visual associations in the post-scan associative test.

**Source data 2.** Percentages of correctly inferred trials during the inference test, split according to performance in the post-scan associative test.

**Source data 3.** Behavioural performance on the inference test versus on the post-scan associative test.

suitable paradigm to investigate the neural mechanisms that support associative recall, in this case for auditory-visual associations.

## Neural signatures of associative memory recall during inference

To investigate neural signatures of associative memory recall during the inference test, we implemented a novel imaging sequence (*Ip et al., 2019*; *Ip et al., 2017*) which enabled interleaved acquisition of near-whole brain fMRI together with fMRS in primary visual cortex (V1) (*Figure 3A*).

The fMRI-fMRS imaging sequence (*Figure 3A*) provided a means to simultaneously measure both haemodynamic and neurochemical changes in an event-related manner. By incorporating a temporal

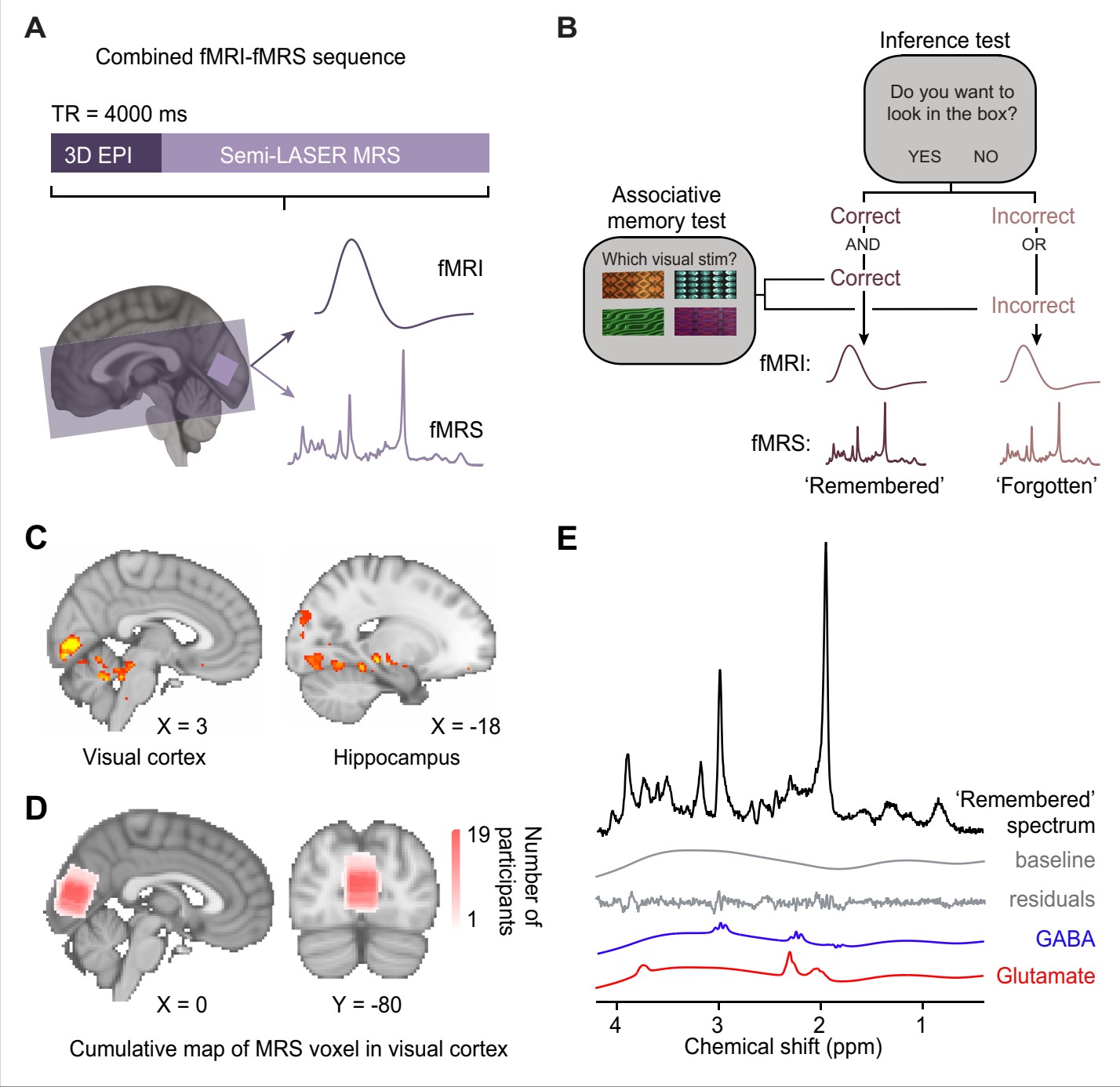

**Figure 3.** Using functional magnetic resonance imaging (fMRI)-functional magnetic resonance spectroscopy (fMRS) data to assess changes in blood oxygen level-dependent (BOLD) signal and glu/GABA ratio during the inference test. (**A**) 7 T MRI sequence. 3D BOLD echo planar imaging (3D-EPI) and semi-LASER MR spectroscopy were acquired in the same TR. The MRS voxel was positioned in primary visual cortex (V1) (light-purple) and the EPI slice coverage included occipital and temporal lobes (dark-purple). (**B**) Schematic showing how trials during the inference test were categorized into 'remembered' and 'forgotten'. Trials were categorised as 'remembered' if participants correctly inferred the appropriate outcome during the inference test *and* subsequently indicated the correct auditory-visual association in the post-scan associative test. Trials were categorised as 'forgotten' if participants incorrectly inferred the appropriate outcome during the inference test *or* indicated the incorrect auditory-visual association in the post-scan associative memory test. Notably, by using performance on the associative test to filter trials on the inference test, this approach helped eliminate false positive and false negative responses during the inference test where chance was otherwise at 50 % (*Figure 2C*). This conservative approach thus provides a more reliable measure of associative memory recall during inference. (**C**) During the question period in the inference test (*Figure 1C–D*), BOLD signal in the visual cortex and the hippocampus was significantly higher for 'remembered' versus 'forgotten' auditory cues ('remembered'–

*Figure 3 continued on next page*

*Figure 3 continued*

'forgotten', visual cortex: $t_{17}$ = 5.92, p < 0.001; hippocampus: $t_{17}$ = 4.33, p = 0.017; whole-volume family wise error (FWE)-corrected; together with regions listed in *Supplementary file 4*; Montreal Neurological Institute [MNI] coordinates). (**D**) Anatomical location of 2 × 2 × 2 cm³ MRS voxel positioned in V1. Cumulative map across participants; MNI coordinates. (**E**) Representative MRS spectrum from 'remembered' trials in the inference test for an example participant.

The online version of this article includes the following source data and figure supplement(s) for figure 3:

**Source data 1.** SPM output for 'remembered' – 'forgotten' contrast.

**Source data 2.** Cumulative map of MRS voxel location across participants.

**Source data 3.** LCModel output for an example 'remembered' MRS spectrum of a single participant.

**Figure supplement 1.** Comparison of different smoothing parameters applied to functional magnetic resonance imaging (fMRI) data.

jitter in each trial of the experimental paradigm (*Figure 1D*), the relationship between data acquisition and the experimental paradigm varied on a trial-by-trial basis (*Figure 4—figure supplement 1*). Therefore, across trials it was possible to effectively assess data at a higher temporal resolution than that given by a TR of 4 s.

In the inference test, participants were required to make a binary 'yes'/'no' response, with chance at 50 %. To exclude trials where participants guessed, we classified trials as 'remembered' or 'forgotten' using a conservative approach. We filtered trials during the inference test post-hoc using participants' behavioural performance from the subsequent post-scan associative test (*Figure 2B*). Trials where participants made both the correct inference (inference test; chance 50%) and indicated the correct auditory-visual associations (associative test; chance 1.6%) were classified as 'remembered'. Trials where participants made either the incorrect inference (inference test) or indicated an incorrect auditory-visual association (associative test) were classified as 'forgotten' (*Figure 3B*, *Supplementary file 2*, Materials and methods). Neural signatures acquired during the 'forgotten' trials thus provided a condition- and stimulus-matched control for data acquired during the 'remembered' trials. Notably, this approach to categorising trials during the inference test controlled for false positives in the inference test, providing a conservative estimate of trials where participants remembered the auditory-visual associations. Notably, there was no significant difference between the number of trials in set 1 (rewarding) versus set 2 (neutral) for the 'remembered' and 'forgotten' conditions (memory × set, two-way ANOVA: $F_{(1,68)}$=0.67, p = 0.424; *Supplementary file 3*).

Using the fMRI data from the interleaved sequence, we first identified brain regions modulated by recall of a visual cue during the inference test (*Figure 1D*). Consistent with previous research investigating associative recall of visual cues (*Horner et al., 2015*; *Wimmer and Shohamy, 2012*) and data acquired using the same task (*Barron et al., 2020*), we observed a significant increase in BOLD signal in both the hippocampus and visual cortex on 'remembered' versus 'forgotten' trials (*Figure 3C*; *Figure 3—figure supplement 1*).

## Dynamic increase in the ratio between glutamate and GABA in visual cortex during recall

We then asked whether associative memory recall of a visual cue is accompanied by changes in the ratio between glutamate and GABA ('glu/GABA ratio', see Materials and methods) in visual cortex. We chose this ROI because recalling a visual cue is known to involve reinstating cortical representations in visual cortex (*Bosch et al., 2014*; *Wheeler et al., 2000*), including during inference as verified with an independent fMRI data set using the same task (*Barron et al., 2020*). Using the interleaved fMRS data acquired in V1 (*Figure 3A and D*), we quantified the concentration of glutamate and GABA normalised to total creatine (tCr) in an event-related manner (*Figure 3B and E*). Notably, to assess dynamic changes in GABA, in the metabolite fitting procedure, it was not appropriate to employ default settings used to detect static estimates of GABA (Appendix 1–supplementary note 1). Importantly, these default settings constrain values of GABA relative to more stable metabolites, a process that effectively limits the dynamic range of GABA (*Figure 4—figure supplement 2*). Instead, here, we use unconstrained GABA estimates (see Materials and methods): while this approach leads to GABA estimates that are higher than values normalised by the concentration of more stable metabolites, critically, dynamic changes in GABA can be detected (*Figure 4—figure supplement 2*).

We used MRS-derived measures of glutamate and GABA to estimate changes in glu/GABA ratio (*Shibata et al., 2017*). During associative memory recall in the inference test, we observed an increase in glu/GABA ratio in V1 when comparing 'remembered' versus 'forgotten' trials (*Figure 4A-C*). Standard quality metrics indicated that our data quality was reliable over the course of the acquisition (*Figure 4—figure supplement 3*, *Supplementary file 5*). To control for any biases introduced by differences in the number of 'remembered' versus 'forgotten' trials (*Supplementary file 6*), we compared the group mean metabolite change against a null distribution generated by permuting the identity labels ('remembered' or 'forgotten') assigned to each trial. This analysis again revealed a significant increase in glu/GABA ratio during memory recall, together with a significant decrease in GABA (*Figure 4D–F*).

These findings cannot be explained by differences in data quality measures between the 'remembered' and 'forgotten' conditions (*Figure 4—figure supplement 4*). In addition, the reported change in glu/GABA ratio was still observed when categorising trials into 'remembered' and 'forgotten' using performance on the inference task alone, a less conservative approach (*Figure 4—figure supplement 5*). The increase in glu/GABA ratio was not observed during periods immediately before or after recall (*Figure 4A–B*; *Figure 4—figure supplement 6*). Moreover, no effect between 'remembered' and 'forgotten' was observed in NAA, a neurometabolite that has overlapping peaks with GABA but is found at higher concentration (*Figure 4—figure supplement 7*). Notably, the observed within-subject, task-specific changes in neurochemistry were obscured when assessing the relationship between average glutamate and average GABA across subjects ($r_{17}$ = 0.191, p = 0.433; after regressing out sex and age: $r_{17}$ = 0.205, p = 0.400), consistent with previous findings (*Rideaux, 2021*). Thus, we propose that the reported transient increase in neocortical glu/GABA ratio reflects a mechanism for associative memory recall.

As an additional control, we assessed changes in glu/GABA ratio during a subset of conditioning trials (*Figure 4—figure supplement 8A*) that were interleaved with the inference test trials during the MRI scan and shared the same temporal structure. Importantly, previous work suggests that performance on conditioning trials is not hippocampal-dependent (*Barron et al., 2020*). During the conditioning trials, we observed no change in glu/GABA ratio during presentation of the visual cue or outcome, relative to the ITI period (*Figure 4—figure supplement 8B, C*).

We note that our MRS sequence does not use editing techniques which exploit known J-coupling relationships to separate signals deriving from low concentration metabolites, such as GABA, from stronger, overlapping signals (*Mullins et al., 2014*). Instead, we implemented an MRS sequence without editing while taking advantage of the benefits associated with using a short TE. To further assess the sensitivity of our approach to detecting dynamic changes in GABA across task conditions, we used Monte Carlo simulations to generate MRS spectra while preserving the observed noise in our data. Using these simulations we show that the observed difference in GABA between 'remembered' and 'forgotten' conditions is significant from a null distribution that would be expected by chance (*Figure 4G*).

## A hippocampal index for fluctuations in neocortical ratio between glutamate and GABA

We next asked which brain regions coordinate this transient break in neocortical glu/GABA ratio during memory recall. The hippocampus is a promising candidate, given this brain region supports memory (*Squire, 1992*) and shows activity modulation during the inference test (*Figure 3C*). To test this possibility, we took advantage of our simultaneous fMRI-fMRS acquisition (*Figure 3A*). We hypothesized that the increase in hippocampal BOLD signal observed during recall (*Figure 3C*) should predict the increase in glu/GABA ratio observed in V1 (*Figure 4B and F*). In line with this prediction, across participants the hippocampal BOLD signal negatively predicted the relative concentration of GABA, and positively predicted the increase in glu/GABA ratio in V1 ('remembered' versus 'forgotten' trials; *Figure 5A–B*). This relationship between the BOLD signal and glu/GABA ratio was not observed in two control regions of interest (ROIs) (*Figure 5—figure supplement 1A, B*). Furthermore, across the imaged brain volume (*Figure 3A*), only the hippocampus significantly predicted the increase in V1 glu/GABA ratio on 'remembered' versus 'forgotten' trials (*Figure 5C*). Finally, this relationship between the hippocampus and glu/GABA ratio was specific to the recall period during the inference test (*Figure 5D*, *Figure 5—figure supplement 1C, D*).

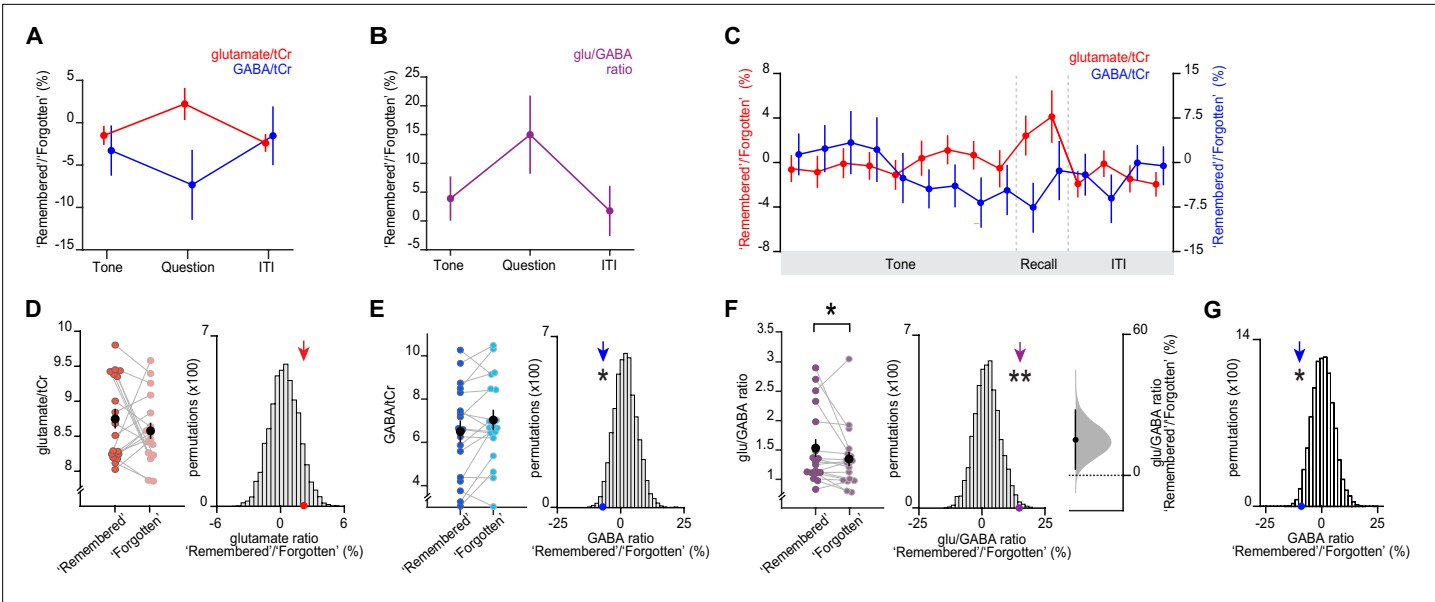

**Figure 4.** Memory recall and inference involves a transient break in glu/GABA ratio. (**A–B**) During the question period of the inference test trials (up to 3 s), glu/GABA ratio significantly increased during 'remembered' versus 'forgotten' trials ('remembered':'forgotten', glu/GABA ratio: $t_{17}$ = 2.21, p = 0.041). One participant (n = 1) was excluded from this analysis, where there were less than eight spectra for either the 'remembered' or 'forgotten' conditions during the question period (see Materials and methods). This break in glu/GABA ratio was not observed during the 'tone' (~7 s) or inter-trial interval ('ITI') (~2.7 s) periods ('tone', glu/GABA ratio: $t_{18}$ = 1.01, p = 0.325; 'ITI', glu/GABA ratio: $t_{18}$ = 0.40, p = 0.692). Note that glutamate:tCR (total creatine) and GABA:tCr concentrations have been multiplied by eight as per convention. To detect dynamic changes in glu/GABA, we chose not to use LCModel's default settings which assume the dynamic range of GABA is fixed (see Materials and methods; *Figure 4—figure supplement 2*, Appendix 1—supplementary note 1). (**C**) Moving average showing glutamate:tCr and GABA:tCr for the ratio of 'remembered' to 'forgotten' trials during the inference test. Each point represents a 2.5 s time bin (mean ± SEM). By incorporating a random jitter in the behavioural paradigm, MRS spectra across all trials and all participants were acquired in all possible 2.5 s time bins of the inference test trial (*Figure 4—figure supplement 1*), thus achieving a higher temporal resolution than the TR of 4 s (see Materials and methods). (**D–F**) Left: The metabolite values and glu/GABA ratio during the question period for 'remembered' and 'forgotten' trials (mean ± SEM). (**D, E**) Right, (**F**) Middle: Comparing the mean ratio of 'remembered' to 'forgotten' (coloured arrows) against null distributions generated by permuting the trial labels to control for any potential biases in the analyses. Relative to the null distributions, GABA significantly decreased while glu/GABA ratio significantly increased (glutamate:tCr: p = 0.089; GABA:tCr: p = 0.014; glu/GABA ratio: p = 0.007). Note: To detect dynamic changes in GABA, it was not appropriate to normalise GABA estimates relative to the concentration of more stable metabolites (see Materials and methods). Consequently, GABA values are higher than those generated using default settings in LCModel which are optimised for detecting static estimates (*Figure 4—figure supplement 2*). (**F**) Right panel: full sampling-error curve for glu/GABA ratio estimated using bootstrap-coupled estimation (DABEST) plot (*Ho et al., 2019*). The 95 % confidence interval is non-overlapping with zero (p = 0.017). Black dot, mean; black tick, 95 % confidence interval; filled-curve, sampling-error distribution. (**G**) The average measured spectra were used as an input to Monte Carlo simulations, to generate simulated spectra with the level of noise matched to the observed data (see *Figure 4—figure supplement 2*). Using this simulated data, we established a null distribution for the difference between pairs of 'remembered' and 'forgotten' spectra that would be expected by chance (i.e. when the condition labels are shuffled). Relative to this null distribution, the observed GABA ratio measured in vivo (shown in E) was significant (p = 0.019). * indicates p < 0.05, ** indicates p < 0.01.

The online version of this article includes the following source data and figure supplement(s) for figure 4:

**Source data 1.** Glutamate:tCr and GABA:tCr 'remembered'/'forgotten' ratios during the 'tone', 'question', and 'ITI' periods of inference trials.

**Source data 2.** Glu/GABA ratio during the 'tone', 'question', and 'ITI' periods of inference trials.

**Source data 3.** Moving average of glutamate:tCr and GABA:tCr for the ratio of 'remembered' to 'forgotten' trials during the inference test.

**Source data 4.** Glutamate:tCr during the question period for 'remembered' and 'forgotten' trials, and null distribution generated by permuting the trial labels.

**Source data 5.** GABA:tCr during the question period for 'remembered' and 'forgotten' trials, and null distribution generated by permuting the trial labels.

**Source data 6.** Glu/GABA ratio during the question period for 'remembered' and 'forgotten' trials, and null distribution generated by permuting the trial labels.

**Source data 7.** Monte Carlo simulated GABA:tCr ratio for 'remembered' vs 'forgotten' trials.

**Figure supplement 1.** Estimation of functional magnetic resonance spectroscopy (fMRS) moving average.

*Figure 4 continued on next page*

*Figure 4 continued*

**Figure supplement 2.** Monte Carlo simulations to assess magnetic resonance spectroscopy (MRS) data quality.

**Figure supplement 3.** Magnetic resonance spectroscopy (MRS) data quality metrics across all spectra.

**Figure supplement 4.** The transient break in glu/GABA ratio observed during recall cannot be explained by changes in data quality metrics or goodness of model fit.

**Figure supplement 5.** An increase in glu/GABA ratio in primary visual cortex (V1) during memory recall is also observed when categorising trials into 'remembered' and 'forgotten' using a less conservative approach.

**Figure supplement 6.** The change in glu/GABA ratio is transient and only observed during memory recall.

**Figure supplement 7.** The changes in metabolite concentrations cannot be attributed to changes in NAA:tCr (total creatine).

**Figure supplement 8.** During conditioning trials, no difference in glu/GABA ratio was observed.

## Discussion

The hippocampus is thought to provide an index for memories stored across distributed neocortical

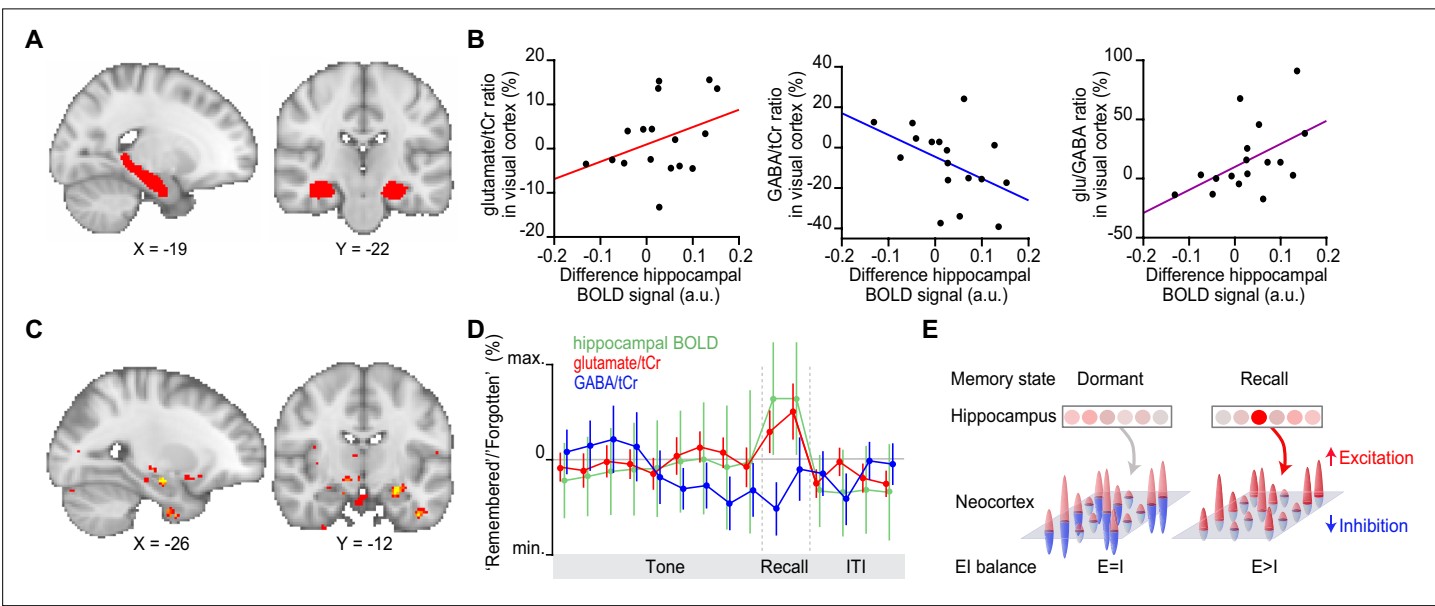

**Figure 5.** Hippocampal blood oxygen level-dependent (BOLD) signal predicts neocortical glu/GABA ratio during recall. (**A**) Region of interest (ROI) in the hippocampus (red). MNI coordinates. (**B**) Across participants, the increase in hippocampal BOLD signal during 'remembered' compared to 'forgotten' trials positively predicted the decrease in GABA and the increase in glu/GABA ratio observed in primary visual cortex (V1) (*Figure 4*) (glutamate:tCr [total creatine]: $r_{15} = 0.14$, $p = 0.585$; GABA:tCr: $r_{15} = -0.56$, $p = 0.022$; glu/GABA ratio: $r_{15} = 0.52$, $p = 0.033$). (**C**) Across the imaged brain volume, for 'remembered' versus 'forgotten' trials, the correlation between the BOLD signal and V1 glu/GABA ratio was selectively observed in the left hippocampus ($t_{16} = 11.37$, $p = 0.005$, whole-brain family wise error (FWE)-corrected; *Supplementary file 7*; MNI coordinates). (**D**) Moving average showing the ratio of 'remembered' to 'forgotten' trials during the inference test: hippocampal BOLD signal (green, range [–4:4]), glutamate:tCr (red, range [–8:8]), GABA:tCr (blue, range [–15:15]). Each point represents a 2.5 s time bin (mean ± SEM). By incorporating a random jitter in the behavioural paradigm, MRS spectra across all trials and all participants were acquired in all possible 2.5 s time bins of the inference test trial (*Figure 4—figure supplement 1*), thus achieving a higher temporal resolution than the TR of 4 s (see Materials and methods). (**E**) Schematic illustrating how the hippocampus may facilitate (if indirectly) memory recall of a sensory cue during a transient break in neocortical EI balance.

The online version of this article includes the following source data and figure supplement(s) for figure 5:

**Source data 1.** Region of interest (ROI) in the hippocampus.

**Source data 2.** Hippocampal BOLD contrast for 'remembered'–'forgotten' versus glutamate:tCr, GABA:tCr and glu/GABA 'remembered'/'forgotten' ratio.

**Source data 3.** SPM output for correlation between 'remembered'–'forgotten' contrast and glu/GABA ratio.

**Source data 4.** Moving average of the ratio of 'remembered' to 'forgotten' trials during the inference test for the hippocampal BOLD signal, glutamate:tCr and GABA:tCr.

**Figure supplement 1.** Before and after memory recall, the hippocampal blood oxygen level-dependent (BOLD) signal did not positively predict glu/GABA ratio in primary visual cortex (V1).

circuits (*Goode et al., 2020*; *Teyler and DiScenna, 1985*; *Teyler and Rudy, 2007*). However, the mechanism by which hippocampal activity coordinates with neocortex to facilitate memory recall has remained unclear. Here, using time-resolved fMRI-fMRS in humans, we show that recall of a visual cue is accompanied by a dynamic increase in the ratio between glutamate and GABA in visual cortex. This transient increase in glu/GABA ratio in visual cortex is selectively predicted by activity in the hippocampus. Accordingly, we propose the hippocampus gates recall of memories stored across distributed neocortical circuits using a disinhibitory mechanism (*Figure 5E*). This mechanism may explain how a memory index represented by the hippocampus selectively releases otherwise dormant representations stored across distributed neocortical circuits.

By simultaneously acquiring both fMRI and fMRS data, we provide a macroscopic readout of memory recall that reflects the consequence of underlying neural circuit level processes. Insight into the nature of these underlying circuit level processes can be gained from related data from animal models. For example, the neural circuit mechanisms that underlie an increase in glu/GABA ratio during recall may be informed by evidence that the ratio between excitatory and inhibitory synaptic conductances in cortical neurons fluctuate around a stable set point (*Anderson et al., 2000*; *Okun and Lampl, 2008*; *Wehr and Zador, 2003*; *Wilent and Contreras, 2005*). This overall EI proportionality ensures that neurons and networks are neither hypo- nor hyper-excitable for prolonged periods, allowing memories to be held in a dormant state (*Barron et al., 2016*; *Froemke et al., 2007*; *Vallentin et al., 2016*; *Vogels et al., 2011*) that is protected from interference caused by new learning (*Koolschijn et al., 2019*; *Kuchibhotla et al., 2017*). However, despite overall proportionality, the exact E/I ratio is highly dynamic and transient breaks in EI balance appear necessary for new learning and memory expression (*Letzkus et al., 2015*). Here, the reported fluctuations in MRS-derived glu/GABA ratio during memory recall may therefore reflect, if indirectly, dynamic changes in EI balance.

Similarly, the reported relationship between the fluctuations in glu/GABA ratio and hippocampal activity may be informed by data from animal models. Of particular relevance are studies in rodents which show that glutamatergic projections from higher-order or interconnected brain regions can target disinhibitory cortical circuits to provide selective EI modulation (*Krabbe et al., 2019*; *Lee et al., 2013*; *Zhang et al., 2014*). For example, to enhance visual discrimination during attentional modulation, projections from the cingulate region of mouse frontal cortex modulate activity in V1 by targeting vasoactive intestinal polypeptide-expressing (VIP+) interneurons, which in turn preferentially target other interneuron subtypes to release excitatory principle cells from inhibitory control (*Zhang et al., 2014*). During memory recall, hippocampal projections may similarly permit memory reinstatement by targeting disinhibitory circuits in neocortex. The correlation between hippocampal activity and glu/GABA ratio reported here may therefore reflect a mechanism whereby activity in the hippocampus facilitates cortical disinhibition to release otherwise latent cortical associations from inhibitory control.

This interpretation of the data is consistent with the notion that the hippocampus provides a memory index to flexibly reinstate information in extrahippocampal circuits (*Goode et al., 2020*; *Teyler and DiScenna, 1985*; *Teyler and Rudy, 2007*). Moreover, our findings replicate equivalent analyses conducted on fMRI data acquired using the same task (*Barron et al., 2020*) and are consistent with previous studies in humans showing evidence for coordinated hippocampal-neocortical memory reinstatement (*Horner et al., 2015*; *Pacheco Estefan et al., 2019*). When combined with the fMRS data, our results also corroborate findings in humans showing that hippocampal glutamate and GABA can predict mnemonic control (*Nikolova et al., 2017*; *Schmitz et al., 2017*). Taken together, we propose a mechanism for hippocampal indexing whereby hippocampal projections control the release of mnemonic representations in sensory cortices by targeting disinhibitory circuits.

Given this interpretation of the data, we emphasise that the relationship between MRS-derived measures of glutamate and GABA and physiological measures of EI balance remains complex. Rapid changes in synaptic glutamate and GABA that accompany neurotransmitter release occur on a timescale that is not possible to detect using fMRS. Moreover, only a fraction of MRS-derived neurometabolite concentration reflects neurotransmitter release. Of the different pools of glutamate and GABA (cytoplasmic, vesicular, or extracellular), MRS is considered most sensitive to unconstrained, intracellular metabolic pools that reside at relatively high concentration in the neuronal cytoplasm (*Rae, 2014*). Indeed, changes in extracellular GABA of less than 100-fold are unlikely to be detectable using MRS (*Myers et al., 2016*) and post-mortem studies suggest MRS is not sensitive to intracellular pools

that reside in the mitochondria or vesicles (*De Graaf and Bovée, 1990*; *Kauppinen and Williams, 1991*).

Interpretation of MRS-derived glutamate and GABA is further complicated by the fact that the release and recycling of glutamate and GABA constitute major metabolic pathways (*Bak et al., 2006*; *Magistretti and Allaman, 2015*). Yet, the metabolic and neurotransmitter pools are thought to be tightly coupled during anaesthesia, rest and certain stimulation protocols, with a 1:1 relationship reported between the rate of glutamine-glutamate cycling, which is necessary for glutamate and GABA synthesis, and neuronal oxidative glucose consumption, which indirectly supports neurotransmitter release among other processes (*Rothman et al., 2003*; *Shen et al., 1999*; *Sibson et al., 1998*). Therefore, an increase in synaptic neurotransmission occurs together with an increase in synthesis of exogenous glutamate, which provides a precursor for GABA via the glutamate-glutamine cycle. During sensory stimulation a transient uncoupling has been observed with a short-lived mismatch between glucose utilization and oxygen consumption (*Fox et al., 1988*; *Fox and Raichle, 1986*), particularly during stimulation protocols that alternate between high intensity and quiescent periods (*Gjedde et al., 2002*). Dynamic fluctuations in fMRS-derived glutamate and GABA reported here may therefore reflect transitions to new metabolic steady states (*Stanley and Raz, 2018*), which could reflect (if indirectly) relative shifts in EI equilibrium at the physiological level. During associative memory recall, the increase in glu/GABA ratio may therefore be interpreted as an increase in synthesis of glutamate relative to degradation, with an opposing effect on GABA.

This interpretation is supported by a handful of previous studies showing event-related changes in MRS glutamate (*Apšvalka et al., 2015*; *Gussew et al., 2010*; *Lally et al., 2014*) and GABA (*Cleve et al., 2015*), together with a growing body of evidence reporting a relationship between MRS-derived measures of neurometabolites and behaviour (*Puts et al., 2011*; *Scholl et al., 2017*; *Stagg et al., 2011*). Nevertheless, it remains to be established whether unconstrained glutamatergic and GABAergic pools show event-related changes that are MRS-sensitive. To validate this interpretation of event-related fMRS, it is important to leverage animal studies where more sensitive methods can be employed to relate fMRS measures to physiological parameters. Here, by implementing an inference task in VR, we operationalize memory recall using the exact same paradigm previously employed in rodents (*Barron et al., 2020*). Therefore, in addition to engaging memory-dependent inference, 'opening the box' to find a reward in the VR environment approximated the process of rodents finding a reward from a dispenser in a 3D environment. By using VR, the findings presented here may be compared to data acquired in animal models in ongoing future research. In this manner, VR paradigms in humans may provide a basis from which to gain insight into the cellular and circuit mechanisms that underlie macroscopic measures of EI. This may prove particularly useful for establishing a more detailed understanding of the relationship between fMRS-derived measures of glutamate and GABA and physiological measures of EI balance.

Previous MRS protocols typically employ a 'block' design, where a static measure of the concentration of glutamate and GABA is achieved by averaging the spectra across a time window that may span several minutes. This approach obscures the temporal dynamics of neurometabolites which more closely relate to fluctuations in EI reported at the physiological level. Similarly, dynamic changes in neurometabolites that accompany cognitive processes and ongoing behaviour are overlooked. Indeed, when the average concentration of Glx and GABA are considered in V1 across time, no significant relationship is observed across subjects (*Rideaux, 2021*), a result which we also observed when assessing average glutamate and GABA using our dataset. By contrast, with the increase in availability of ultra-high field MRI scanners and the development of more advanced sequences (*Stagg and Rothman, 2013*), fMRS has emerged as a viable method to detect dynamic changes in neurochemicals in both healthy and clinical populations (*Stanley and Raz, 2018*).

Although there are currently only a handful of event-related fMRS studies, together with our data, these suggest that fMRS is highly sensitive to detecting task-relevant dynamic changes in glutamate and GABA (*Jelen et al., 2018*). For example, in the lateral occipital complex, fMRS demonstrates differences in glutamate in response to presentation of objects versus abstract stimuli (*Lally et al., 2014*), and in the left anterior insula fMRS reveals a transient increase in glutamate with exposure to painful stimuli (*Gussew et al., 2010*). fMRS-derived glutamate is even sufficiently sensitive to detect repetition suppression effects in the lateral occipital complex (*Apšvalka et al., 2015*), mirroring analogous effects reported in fMRI (*Barron et al., 2016*; *Grill-Spector et al., 2006*). Here, we further illustrate

that within a 3 s time window delineated by the question period in the inference task, the temporal resolution of fMRS is sufficient to relate transient changes in glutamate and GABA to associative memory recall. Importantly, we compare data across two conditions ('remembered' and 'forgotten') to inherently control for: (1) between-subject differences in average GABA and glutamate which are affected by demographic (e.g. age and gender); (2) between-subject differences in spectral quality; (3) between-subject differences in tissue composition; (4) between-subject differences in the effect of other neurochemicals on measures of glutamate and GABA. Such time-resolved, within-subject, and condition-dependent fMRS may provide a promising tool to capture real-time, task-relevant changes in neurometabolites, on a timescale equivalent to task-based fMRI. Assessing whether the temporal resolution of fMRS can be further improved will likely prove an important step in refining fMRS in the future.

During associative memory recall, the transient increase in glu/GABA ratio reported in our data can be accounted for by a significant decrease in the concentration of MRS-derived GABA. Notably, detecting dynamic changes in GABA is challenging for two key reasons: the concentration of GABA in human brain tissue is relatively low and the spectral peaks for GABA overlap with other, more abundant neurochemicals (*Andreychenko et al., 2012*; *Govindaraju et al., 2000*; *Puts and Edden, 2012*). While the most common approach to detecting MRS-derived GABA involves using a J-difference spectral editing technique to separate GABA peaks from overlapping peaks (*Bottomley, 1987*; *Mescher et al., 1998*), here we use a non-edited sequence (sLASER). While spectral editing may provide higher precision (*Hong et al., 2019*), this occurs at the cost of a larger volume of interest, longer TEs and higher susceptibility to motion and drift artefacts due to longer acquisition times, making it less suitable for event-related fMRS (*Terpstra et al., 2006*; *Trabesinger and Boesiger, 2001*). Moreover, direct comparisons between edited and non-edited sequences at 7 T reveal no significant difference in the concentration of GABA measurements (*Hong et al., 2019*). Therefore, together with studies reporting dynamic changes in GABA with sensory stimulation (*Lin et al., 2012*; *Mekle et al., 2017*), our data illustrates how a non-edited sequence can provide sufficient data quality for measuring dynamic changes in MRS-derived GABA, which cannot be explained by changes in compounds at higher concentration that have overlapping peaks (i.e. glutamate or NAA, *Figure 4—figure supplement 7*). Indeed, Monte Carlo simulations reported here validate that non-edited sequences can be used to quantify dynamic changes in GABA (*Figure 4G*; *Figure 4—figure supplement 2*). Moreover, compared to spectral editing, our approach comes with the advantage of simultaneously measuring dynamic changes in GABA and glutamate, together with 17 other neurometabolites.

To detect dynamic changes in GABA, it was necessary to disable default priors on the spectral fitting procedure that constrain GABA as a ratio to more stable metabolite concentrations (*Figure 4—figure supplement 2*, see Appendix 1—supplementary note 1). As a consequence, we were able to detect dynamic changes in both glutamate and GABA across time, as illustrated using Monte Carlo simulations and permutation testing. By comparing the change in metabolite concentration between two conditions ('remembered' versus 'forgotten'), the ratio in GABA between conditions rather than absolute values was the key measure of interest. However, we note that absolute GABA estimates were higher compared to those obtained using default priors that normalise estimates relative to more stable metabolite concentrations. Importantly, the quality of our MRS data was comparable with other studies that have acquired 7 T MRS data from visual cortex (*Bednařík et al., 2018*; *Hong et al., 2019*; *Mekle et al., 2017*; *Prinsen et al., 2017*). Moreover, the quality of the glutamate estimates was in line with previous studies employing event-related fMRS to assess dynamic changes in glutamate (*Apšvalka et al., 2015*; *Gussew et al., 2010*; *Lally et al., 2014*).

Disturbances in EI balance are thought to underlie a number of neuropsychiatric conditions, including schizophrenia, autism, epilepsy, and Tourette's syndrome (*Robertson et al., 2016*; *Stanley and Raz, 2018*; *Taylor et al., 2015*). While previous studies report inconsistencies in MRS-derived measures of glutamate and GABA in these patient populations, this may be attributed to differences in brain region, cognitive state, and imaging protocol, among other factors. Here, by using both fMRS and fMRI to reveal a signature change in glu/GABA ratio that relates to hippocampal BOLD signal, behavioural performance, and cognition, our findings present a potential target for clinical investigation. Moreover, our findings show that even in the healthy brain a transient break in EI balance is necessary to support key cognitive processes such as memory recall.

In summary, using time-resolved fMRI-fMRS, we report a transient increase in glu/GABA ratio in V1 during associative recall of a visual cue. This increase in neocortical glu/GABA ratio is predicted by activity in the hippocampus. By unveiling this coordination between the hippocampus and neocortex, we show how the hippocampus may have the capacity to selectively modulate and disinhibit memories represented in neocortex. This mechanism may explain how the hippocampus plays a key role in memory recall, by indexing the release of specific memories stored across distributed neocortical circuits.

## Materials and methods

### Resource availability

#### Lead contact

Further information and requests for resources should be directed to and will be fulfilled by the lead contact, Helen Barron (helen.barron@merton.ox.ac.uk).

### Materials availability

This study did not generate new unique reagents.

### Experimental model and subject details

#### Participants

Twenty-two healthy human volunteers were included in the study (mean age of 22.8 ± 0.74 years, four males). All experiments were approved by the University of Oxford ethics committee (reference number R43594/RE001). All participants gave informed written consent. For one participant, we were unable to collect combined fMRI-fMRS data due to time constraints during scanning. Two participants were excluded from the fMRI and fMRS analyses due to technical difficulties which resulted in the auditory cues not being fully audible during the inference test. Notably, there was no significant effect of sex on either behavioural performance or MRS measures of glu/GABA ratio during the inference test (*Supplementary file 1*).

### Method details

#### VR environment

The VR environment was coded using Unity 5.5.4f1 software (Unity Technologies, San Francisco, CA) (*Figure 1A*). The VR environment was designed to simulate an open field environment previously used to investigate memory and inference in mice (*Barron et al., 2020*). Within the VR environment, participants were exposed to a range of different sensory stimuli, in accordance with the three-stage inference task described below.

The environment included a square-walled room with no roof (*Figure 1A*). To help evoke the experience of 3D space and aid orientation within the VR environment, each wall of the environment was distinguished by colour (dark green, light green, dark grey, or light grey), illumination (two walls were illuminated while the other two were in shadow), and by the presence of permanent visual cues. The permanent visual cues included clouds in the sky, a vertical black stripe in the middle of the light green wall, a horizontal black strip across the light grey wall, and a wooden box situated in one corner of the environment. A first-person perspective was implemented and participants could control their movement through the virtual space using the keyboard arrows (2D translational motion) and the mouse-pad (head tilt). Movement through the environment elicited the sound of footsteps. Within the VR environment, participants were exposed to a range of different sensory stimuli, in accordance with the three-stage inference task described below.

#### Three-stage inference task

In the VR environment (*Figure 1A*) humans performed an inference task (*Figure 1B*). The rationale for using an inference task to assess mechanisms responsible for associative memory was threefold. First, evidence in both humans (*Barron et al., 2020*; *Koster et al., 2018*) and mice (*Barron et al., 2020*) shows that performance on this inference task requires associative memory recall. Second, in rodents, inference, but not first-order associative recall, is hippocampal-dependent (*Barron et al.,*

*2020*; *Bunsey and Eichenbaum, 1996*; *DeVito et al., 2010*), thus providing an opportunity to investigate hippocampal-dependent associative memory recall. Third, the task can be deployed across humans and rodents, which may allow future investigation of the cellular mechanisms that underlie non-invasive measures reported here.

The task was adapted from associative inference and sensory preconditioning tasks described elsewhere (*Barron et al., 2020*; *Brogden, 1939*; *Preston and Eichenbaum, 2013*) and involved three stages performed across 3 consecutive days, respectively (*Figure 1B and C*). The first and second stages were performed outside the scanner while the third stage was performed inside the scanner (*Figure 1C*). At the start of the experiment the pairings between auditory, visual, and outcome cues were randomly assigned for each participant.

On day 1, participants performed the 'associative learning' stage (*Figure 1B*), during which participants were required to learn at least 40 (out of 80 total) auditory-visual associations via mere exposure. In total, there were four visual cues, each associated with 20 different auditory cues. Auditory cues constituted 80 different complex sounds (e.g. natural sounds or those produced by musical instruments) that were played over headphones. Visual cues constituted four different unique patterned panels which could appear on the walls of the environment (*Figure 1A and B*). To control for potential spatial confounds, two of the visual cues were always presented on the same wall, the assignment of which was randomized for each participant. The two remaining visual cues were 'nomadic', meaning that with each presentation they were randomly assigned to one of the four walls.

Training during the associative learning stage occurred within the VR environment as described previously (*Barron et al., 2020*). In brief, on each trial of the 'associative learning', an auditory and visual cue were presented serially and contiguously: 8 s auditory cue followed by 8 s of the associated visual cue, followed by an ITI of 5 s (*Figure 1—figure supplement 1A*). Learning of auditory-visual associations was monitored outside the VR environment, using an associative learning test coded in MATLAB 2016b using Psychtoolbox (version 3.0.13). On each trial of the associative learning test, one auditory cue was presented, followed by presentation of four different visual cues (*Figure 1—figure supplement 1B*). Participants were instructed to select the visual cue associated with the auditory cue using a button press response within 3 s, and only at the end of the test were participants given feedback on their average performance. Training on the associative learning stage was terminated only when participants reached >50% accuracy on the associative learning test when all 80 auditory cues were included, each presented three times (*Figure 1—figure supplement 1E*). Those participants that failed to reach >50% accuracy (n = 3) did not proceed to day 2 and were thus not included in the experiment.

On day 2, participants performed the 'conditioning' stage (*Figure 1B*), during which they learned that two of the four visual cues (set 1) predicted delivery of a rewarding outcome (virtual silver coin, as above) on 80 % of trials, while the other two visual cues (set 2) predicted delivery of a neutral outcome (virtual woodchip, as above) on 100 % of trials. Training during the conditioning stage occurred within the VR environment and on each trial, participants were presented with a visual cue (8 s) followed by outcome delivery to a wooden box (available for 6 s) situated in the corner of the environment (*Figure 1—figure supplement 1C*). The inter-trial interval (ITI) was 2 s. To harvest the value of a virtual silver coin (monetary reward later converted to 20 pence per coin) or woodchip (no value, 0 pence), participants were required to collect the coin or woodchip from the wooden box. The cumulative total value of harvested reward was displayed in the upper left corner of the computer screen.

Learning during conditioning was monitored using a conditioning test coded in MATLAB 2016b using Psychtoolbox (version 3.0.13). On each trial of the conditioning test, participants were presented with a still image of a visual cue before being asked to indicate the probability of reward using a number line (*Figure 1—figure supplement 1D*). Participants were given 3 s to respond and were only given feedback on their average performance at the end of the test. Participants were required to repeat the VR conditioning training and conditioning test until they performed the test with 100 % accuracy (*Figure 1—figure supplement 1F*).

Finally, on day 3, participants first repeated the conditioning test. Participants then entered the 7 T MRI scanner and performed the 'inference test' (*Figure 1B–D*), together with a subset of conditioning trials (*Figure 4—figure supplement 8A*) (see *fMRI-fMRS scan task* below). Immediately after exiting the scanner, participants were given a surprise associative test to assess which auditory-visual associations they remembered (*Figure 2A*). The associative test was equivalent to the test performed on day

1 during the associative learning (*Figure 1—figure supplement 1B*), with three trials for each auditory stimulus. Performance on auditory-visual associations was categorised as correct if participants scored 3/3 for a given auditory-visual pair. Performance on auditory-visual associations was categorised as incorrect if participants scored 0/3 or 1/3 for a given auditory-visual pair (i.e. no different from chance). Trials where participants scored 2/3 were not categorised as either correct or incorrect due to their ambiguity. The behavioural performance measured on the post-scan associative test (*Figure 2A*) was a more sensitive measure of memory accuracy than behavioural performance measured during the inference test alone, with a lower chance level (associative test: four choice options with 1.6 % chance level for correct response across three repeats; inference test: two options with 50 % chance level for correct response across one repeat). For this reason, performance on the inference test during the scan was assessed post hoc using performance from both the inference test and the post-scan associative test (see *Trial categorisation during the inference test*, *Figure 3B*).

## fMRI-fMRS scan task

The inference test was incorporated into the fMRI-fMRS scan task. This provided an opportunity to measure neural responses to associative memory recall required for inferential judgements. The scan task included two different trial types: inference test trials (*Figure 1D*) and conditioning trials (*Figure 4—figure supplement 8A*). For both types of trial, participants viewed a short video taken from the VR training environment (*Figure 1—video 1*). The videos were presented via a computer monitor and projected onto a screen inside the scanner bore. On each trial the duration of the video was determined using a truncated gamma distribution with mean of 7 s, minimum of 4 s, and maximum of 14 s. During the inference test trials, the video of the VR environment was accompanied by an auditory cue, played over MR compatible headphones (S14 inset earphones, Sensimetrics). Visual cues were not displayed during these trials: the auditory cues were presented in isolation. At the end of the video, participants were presented with a question asking: 'Would you like to look in the box?', with the options 'yes' or 'no' (*Figure 1D*). Importantly, as described above, outcomes (rewarding or neutral) were delivered to the wooden box during the conditioning stage. Participants were required to make a response within 3 s using an MR compatible button box and their right index or middle fingers. No feedback was given. To infer the appropriate outcome, participants were instructed to use the learned structure of the task. After each trial (inference or conditioning), a cross was presented in the centre of the screen during an ITI of varying length, determined using a truncated gamma distribution (mean of 2.7 s, minimum of 1.4 s, maximum of 10 s). Trials were categorised as 'correctly inferred' if participants pressed 'yes' in response to auditory cues indirectly associated with a rewarding outcome, or pressed 'no' in response to auditory cues indirectly associated with a neutral outcome (*Figure 1E*). The inference test provided an opportunity to investigate memory recall: to infer the correct outcome participants needed to recall the appropriate visual cue associated with the auditory cue (*Figure 2C–D*).

Conditioning trials were interleaved with inference test trials to minimise extinction effects. During conditioning trials, the video of the VR environment orientated towards a visual stimulus displayed on one of the four walls (*Figure 4—figure supplement 8A*). At the end of the video, participants were presented with a still image of the associated outcome for that visual cue (*Figure 4—figure supplement 8A*).

To control for potential confounding effects of space, each video during the inference test involved a trajectory constrained to a 1/16 quadrant of the VR environment, evenly distributed across the different auditory cues. Across conditioning trials, each visual cue was presented 16 times, once in each possible spatial quadrant. Moreover, the videos were not related to the background for the relevant visual cue. Allocation of the videos to each trial was randomised separately for each participant to ensure no consistent biases. The fMRI-fMRS scan task was evenly divided across two scan blocks, each of which lasted 15 min. The fMRI-fMRS scan task was then repeated (two more scan blocks) using a higher quality multiband fMRI sequence (reported elsewhere; *Barron et al., 2020*).

## fMRI-fMRS data acquisition

The fMRI-fMRS scan task was performed inside a 7 T Magnetom MRI scanner (Siemens) using a 1-channel transmit and a 32-channel receive phased-array head coil (Nova Medical Inc, Wilmington, MA) at the Wellcome Centre for Integrative Neuroimaging (University of Oxford). Current 7 T radio frequency coil designs suffer from $B_1^+$ inhomogeneity. To overcome this, we positioned two 110 ×

$110 \times 5$ mm$^3$ Barium Titanate dielectric pads (4:1 ratio of BaTiO$_3$:D$_2$O, relative permittivity around 300) over occipital lobe, causing a 'hotspot' in the proximal B$_1^+$ distribution at the expense of distal regions (*Brink and Webb, 2014*). For each participant, a T1-weighted structural image was acquired to inform placement of the MRS voxel in visual cortex, and to correct for geometric distortions and perform co-registration between EPIs, consisting of 176 0.7 mm axial slices, in-plane resolution of 0.7 $\times$ 0.7 mm$^2$, TR = 2.2 s, TE = 2.96 ms, and field of view = 224 mm. For each participant, a field map with dual echo-time images was also acquired (TE1 = 4.08 ms, TE2 = 5.1 ms, whole-brain coverage, voxel size $2 \times 2 \times 2$ mm$^3$).

*Figure 3A* shows a diagram of the combined fMRI-fMRS sequence, based on a sequence developed by *Hess et al., 2011*, and previously used to compare the BOLD signal in V1 with measures of glutamate (*Ip et al., 2019*; *Ip et al., 2017*). In the same TR of 4.105 s, BOLD-fMRI (3D EPI, resolution $2.3 \times 2.3 \times 2.2$ mm$^3$; flip angle = 5°, repetition time TR$_{epi}$ = 59 ms, TE = 29 ms, field of view 200 mm, 32 slices) and fMRS data ($2 \times 2 \times 2$ cm$^3$ voxel positioned in the occipital lobe, centred along the midline and the calcarine sulcus) were acquired. The TR was increased for four participants where a higher transmit voltage was required, resulting in a TR between 4.7 and 5.9 s. fMRS data were acquired using short-echo-time semi-localisation by adiabatic selective refocusing (semi-LASER) pulse sequence (TE = 36 ms, TR$_{mrs}$ = 4 s) with VAPOR water suppression and outer volume suppression (*Oz and Tkáč, 2011*). A delay between fMRI and fMRS acquisition (250 ms) was inserted to minimize potential eddy current effects from the EPI read-out (*Hess et al., 2011*). Compared to an uncombined contemporary MR sequences (e.g. multiband EPI and semi-LASER MRS), the fMRS was of comparable quality, while the quality of the fMRI component was compromised. The quality of MRS data was assessed during set-up scans and acquisition. Several criteria were considered: we monitored the level of noise in the data and the residual water signal to check for stability and to ensure that the size of the water peak was well below the level of most observable metabolites. During set-up scans, the position of the MRS voxel was adjusted if necessary. Data quality was further assessed during data analysis, using the metrics reported in *Figure 4—figure supplement 3* and listed in *Supplementary file 5*. On average, 457 fMRS spectra were acquired over the two scanning blocks (SD: 35.62).

In addition to the fMRI-fMRS sequence acquisition, an additional set of fMRI data (reported elsewhere [*Barron et al., 2020*] and not shown here) was acquired using a multiband EPI sequence (50 1.5 mm thick transverse slices with 1.5 mm gap, in-plane resolution of $1.5 \times 1.5$ mm$^2$, TR = 1.512 s, TE = 20 ms, flip angle = 85°, field of view 192 mm, and multi-band acceleration factor of 2). To increase SNR in brain regions for which we had prior hypotheses, both the fMRI sequences were restricted to partial brain coverage (*Figure 3A*, covering the occipital and temporal lobes) to shorten the EPI TR, thus acquiring more measurements.

## Trial categorisation during the inference test

Trials during the inference test were categorised into two conditions, 'remembered' and 'forgotten' (*Figure 3B*). Given participants were required to make a binary 'yes'/'no' response in the inference test, chance was at 50 %. To ensure our findings could not merely be explained by false positives, we categorized trials during the inference test into 'remembered' and 'forgotten' using a conservative approach, to ensure we could confidently identify when the associated visual cue was recalled in response to the auditory cue. To this end, we controlled for false positives in the inference test by filtering behavioural performance on the inference test using behavioural performance recorded on the post-scan associative memory test (where correct recall of auditory-visual associations has chance level of 1.6%). Thus, trials where participants made both the correct inference during the inference test *and* indicated the correct auditory-visual association during the post-scan associative test were classified as 'remembered'. Trials where participants made *either* the incorrect inference during the inference test *or* the incorrect auditory-visual association during the post-scan associative test were classified as 'forgotten'. Notably, this approach effectively eliminates trials where participants were guessing during the inference test, thus providing a conservative estimate of trials where the subjects remember the auditory-visual associations.

## Quantification and statistical analysis

### fMRS metabolite quantification and analysis

For each scan run, fMRS data from 19 subjects was preprocessed separately in MRspa, a semi-automated MATLAB routine (https://www.cmrr.umn.edu/downloads/mrspa/). The unsuppressed water signal acquired from the same VOI was used to remove residual eddy current effects and combine individual coil spectra. Spectra were corrected for frequency and phase variations induced by participants' motion, and the residual water component was removed using Hankel Lanczos singular value decomposition (HLSVD). For each participant, spectra from all blocks were frequency aligned to account for frequency differences between blocks. The parameters implemented for fMRS acquisition are summarized in *Supplementary file 8*, according to standards proposed by the 'MRS Experts Working Group' (*Lin et al., 2021*).

Spectra were then analysed in an event-related manner. For each participant, the preprocessed spectra were first assigned to the tone/question/ITI periods by aligning the time stamps for the spectra to the time stamps for each event recorded during the inference task. Then, spectra acquired within the tone/question/ITI periods were selected for analysis. Next, these selected spectra were separated into two categories according to task performance, 'remembered' or 'forgotten' (*Figure 3B*, see *Trial categorisation during the inference test*), before being analysed using LCModel. Participants (n = 1 for the 'Question' period only) with less than eight spectra for either the 'remembered' or 'forgotten' conditions were excluded from the fMRS analysis. Notably, previous studies report minimal change in test-retest CoVs when going from 8 to 16 spectra (*Terpstra et al., 2016*). Metabolite concentrations for the average 'remembered' and the average 'forgotten' spectrum were quantified in turn using LCModel (*Provencher, 1993*) within the chemical shift range 0.5–4.2 ppm. The concentration of each metabolite was assessed relative to the concentration of tCr (creatine + phosphocreatine, tCr), thus providing effective control for variation in voxel tissue and cerebral spinal fluid (CSF) in the fMRS voxel used across participants. Within LCModel, metabolite estimates were not constrained by priors that assume the GABA estimate remains fixed relative to more abundant neurochemicals (i.e. we set the parameter NRATIO to 0, see Appendix 1—supplementary note 1). As a result, relative to default settings, the GABA estimates were higher and the dynamic range of GABA was not assumed to be fixed (*Figure 4—figure supplement 2*). Estimates were normalised to tCr and multiplied by 8, as per convention. A basis set containing stimulated model spectra of alanine (Ala), aspartate (Asp), ascorbate/vitamin C (Asc), glycerophosphocholine (GPC), phosphocholine (PCho), creatine (Cr), phosphocreatine (PCr), GABA, glucose (Glc), glutamine (Gln), glutamate (Glu), glutathione (GSH), myo-inositol (myo-Ins), lactate, *N*-acetylaspartate (NAA), *N*-acetylaspartylglutamate (NAAG), phosphoethanolamine (PE), scyllo-inositol (scyllo-Ins), taurine (Tau), and experimentally measured macromolecules was used.

Changes in the relative concentration of glutamate and GABA between 'remembered' and 'forgotten' conditions were evaluated together with 'glu/GABA ratio' which we defined as the ratio of glutamate to GABA (*Shibata et al., 2017*). We defined the change in glutamate, GABA, and glu/GABA for 'remembered' versus 'forgotten' trials as a ratio, as follows:

$$\text{Glutamateratio} = 100 \times \left( \frac{\text{Glu}_{\text{remem}} - \text{Glu}_{\text{forgot}}}{\text{Glu}_{\text{forgot}}} \right)$$

$$\text{GABAratio} = 100 \times \left( \frac{\text{GABA}_{\text{remem}} - \text{GABA}_{\text{forgot}}}{\text{GABA}_{\text{forgot}}} \right)$$

$$\text{glu/GABAratio} = 100 \times \left( \frac{\frac{\text{Glu}_{\text{remem}}}{\text{GABA}_{\text{remem}}} - \frac{\text{Glu}_{\text{forgot}}}{\text{GABA}_{\text{forgot}}}}{\frac{\text{Glu}_{\text{forgot}}}{\text{GABA}_{\text{forgot}}}} \right)$$

where Glu and GABA represent the ratio of glutamate and GABA to tCr, respectively, during the tone/question/ITI period of 'remembered' or 'forgotten' trials. This ratio effectively controls for variation in voxel tissue and CSF fraction in the MRS voxel used across participants.

Further, to control for differences in the number of 'remembered' and 'forgotten' spectra, we compared the group mean difference between 'remembered' and 'forgotten' trials against a null distribution generated by permuting the trial labels while preserving differences in number of trials for each participant. On each of 5000 permutations, the condition labels ('remembered', 'forgotten') were shuffled for each participant using MATLAB's random number generator. The relative metabolite

concentrations for each condition were then estimated in LCModel and the difference between conditions computed. The group mean for each permutation was then added to the null distribution. The difference between 'remembered' and 'forgotten' conditions derived from the unshuffled data was then compared against the null distribution generated from the shuffled data (*Figure 4D–F*; *Figure 4—figure supplements 4–7*).

To control for a false-positive result and to provide further evidence for the reported change in glu/GABA ratio between 'remembered' and 'forgotten' trials, we generated a sampling-error distribution computed from 10,000 bootstrapped resamples of glu/GABA ratio (*Efron, 2000*). We visualised the effect size of the relative measure using a Data Analysis with Bootstrap-coupled ESTimation (DABEST) plot (*Ho et al., 2019*; *Figure 4F*).

## MRS Monte Carlo simulations

To assess the relative sensitivity to detecting changes in GABA in our data, we used Monte Carlo simulations to generate synthetic spectra. The average observed spectrum (across participants) was used as an input to Monte Carlo simulations (*Clarke et al., 2021*). For each set of conditions, we generated 2000 simulated spectra, with the SNR and line width of the simulated data matched to the SNR and line width observed in the data. The T2 values are assumed to be the same between conditions since we did not see any differences in FWHM between 'remembered' and 'forgotten' conditions in the in vivo data (*Figure 4—figure supplement 4B*). The output of the simulations was then analysed in LCModel to quantify GABA and glutamate.

These simulated spectra were used for two types of analyses. First, the simulated spectra were used to test the likelihood of observing the measured change in GABA between 'remembered' and 'forgotten' conditions by chance (*Figure 4G*). Second, the simulated spectra were used to assess the effect of synthetically imposing changes in GABA, both with and without constraining GABA relative to the concentration of other more abundant neurochemicals ('constraints on' versus 'constraints off') (*Figure 4—figure supplement 2*). The imposed changes in GABA were the following multiples of the observed difference in GABA (between 'remembered' and 'forgotten'): 0, ±0.5, ±1, ±2. To assess the sensitivity of the 'constraints on' and 'constraints off' LCModel settings to changes in imposed GABA at different SNRs, we compared the slope for each setting. Slopes were determined by fitting a general linear model (GLM) to the imposed versus measured GABA concentration in the simulated data. To quantify the difference in slope between categories (*Figure 4—figure supplement 2C*: 'constraints on' versus 'constraints off'; *Figure 4—figure supplement 2D*: differences in SNR), we randomly sampled n = 18 simulated spectra for each imposed change in GABA, for each condition of interest (e.g. 'constraints on' versus 'constraints off'; SNR 125 % versus SNR 75 %, etc.). Over 500 sets of size n = 126 (i.e. n = 18 simulated spectra for each imposed change in GABA), we estimated the power to reject the null hypothesis of equal slopes between conditions. Similarly, using a parametric statistical approach, over 500 sets of size n = 126 simulated spectra, we estimated the t-statistic to reject the null hypothesis of equal slopes between conditions.

## fMRI preprocessing and GLMs

Preprocessing of MRI data was carried out using SPM12 (http://www.fil.ion.ucl.ac.uk/spm/). First, the anterior commissure was set to the origin in the anatomical images and in the first volume of each fMRI block, with equivalent transformations applied to all other images within the same block. Second, to account for magnetic field inhomogeneities, images were corrected for signal bias, realigned to the first volume, corrected for distortion using field maps, normalised to a standard EPI template. To remove low-frequency noise from the preprocessed data, a high-pass filter was applied to the data using SPM12's default settings. For each participant and for each scanning block, the resulting fMRI data was analysed in an event-related manner using a GLM. The GLM was applied to data from both scan task blocks. In addition to the explanatory variables (EVs) of interest (described below), six additional scan-to-scan motion parameters produced during realignment were included in the GLM as nuisance regressors to account for motion-related artefacts in each task block. The output of the first-level analysis was then smoothed using a 5 mm full width at half maximum Gaussian kernel before being entered into a second-level analysis. The sensitivity of our analysis pipeline to detecting stimulus evoked BOLD activity patterns benefitted from applying the first-level GLM to unsmoothed data and only including smoothing prior to the second-level analysis (*Figure 3—figure supplement 1*).

Across participants, data quality was assessed after each step in the preprocessing of the data. One participant was excluded from the fMRI analyses as co-registration between the epi and structural scans failed due to insufficient quality of fMRI data. Notably, this participant was not the same as the participant excluded from the fMRS analyses, where less than eight spectra in either the 'remembered' or 'forgotten' conditions occurred for n = 1 participants (see above).

For the first-level analyses, three different GLMs were used. Each GLM included 15 EVs per block. In the first GLM, the first eight EVs accounted for the question period in the inference test, divided according to performance of the subject ('remembered' or 'forgotten', see *Trial categorisation during the inference test*), before being further divided according to the four possible visual cues to which the auditory cues were associated. The next four EVs accounted for presentation of the visual cue during the video of all conditioning trials, divided according to the four different visual cues. The final three EVs accounted for presentation of the auditory cue during the video in all inference test trials, the question period in all remaining inference test trials (i.e. trials not categorized as 'remembered' or 'forgotten'), and the presentation of the outcome in all conditioning trials. To decorrelate the EVs modelling the auditory and visual cues from those EVs modelling the question and outcome, respectively, the duration of events within EVs modelling the auditory and visual cues was set using a box-car function to 4 s, that is, the minimum duration of the video. The duration of events within EVs modelling the question/outcome were set to the duration of the question/outcome. All EVs were then convolved with the haemodynamic response function.

In the second and third GLMs, the same EVs were included, however, the first eight EVs accounted for the auditory cue period in the inference test (second GLM), or the ITI in the inference test (third GLM). In both cases, the EVs were divided according to performance of the subject ('remembered' or 'forgotten'), as in the first GLM.

## Univariate fMRI analysis and statistics

Using the output of the GLMs, we assessed the difference in the univariate BOLD response between 'remembered' and 'forgotten' trials during the inference test (as defined in *Figure 3B*, *Trial categorisation during the inference test*). The contrast of interest therefore involved contrasting EVs [1:4] ('remembered') with EVs [5:8] ('forgotten'), using the first GLM (see above). The resulting contrast images ('remembered'–'forgotten') for all participants were entered into a second-level random effects 'group' analysis. We set the cluster-defining threshold to $p < 0.01$ uncorrected before using whole-brain family wise error (FWE) to correct for multiple comparisons, with the significance level defined as $p < 0.05$ (*Figure 3C*, *Supplementary file 4*).

## Assessing the relationship between fMRI and fMRS

To assess the relationship between event-related hippocampal BOLD signal and event-related fMRS measures from V1, we used an anatomical ROI for the hippocampus (*Figure 5A*). Capitalising on variance across participants, the relationship between the BOLD signal for 'remembered'–'forgotten' within this ROI was compared with equivalent changes in glutamate, GABA, and glu/GABA ratio using a Spearman rank correlation. To assess the selectivity of these effects to the recall period (question) during the inference test, control analyses were performed using the output of the second and third GLMs, together with equivalent measures of glutamate, GABA, and glu/GABA ratio (*Figure 5—figure supplement 1*).

Next, to assess the relationship between fMRS and the BOLD signal across the entire imaged brain volume, we repeated the second-level random effects 'group' analysis using the output of the first GLM, but now included group-level covariates for the change in glutamate and GABA for 'remembered'–'forgotten' (i.e. *Figure 4A*), along with two 'nuisance' regressors that accounted for unwanted variance attributed to differences in age and sex. To identify brain regions where the BOLD signal for 'remembered':'forgotten' predicted changes in glu/GABA ratio, we contrasted the EVs on the covariates for glutamate and GABA (glutamate–GABA) to generate a single contrast to test statistical significance. We set the cluster-defining threshold to $p < 0.01$ uncorrected before using whole-brain FWE to correct for multiple comparisons, with the significance level defined as $p < 0.05$ (*Figure 5C*, *Supplementary file 7*).

To visualize the time course of fMRS across the inference test trials, we estimated a moving average, where each time bin constituted a 2.5 s time window shifted by 0.5 s in each iteration (*Figures 4C*

*and 5D, Figure 5—figure supplement 1A, B*). By incorporating a random jitter in each trial of the fMRI-fMRS scan task, the temporal relationship between MRS spectra acquisition and the inference test trials varied. Thus, across participants and across trials, MRS spectra were acquired in all possible 2.5 s time bins of the inference test trial, achieving a higher temporal resolution than the TR of 4 s (*Figure 4—figure supplement 1*). To ensure each time bin contained a similar number of spectra, those bins at the tail end of the jitter (final three time bins during the video and the final two time bins of the ITI) were enlarged to include broader time windows ( > 2.5 s). For each participant, the 'remembered' and 'forgotten' spectra were then calculated for each time bin, and the ratio estimated to give a measure of 'remembered':'forgotten' for both glutamate and GABA. For each time bin, data for a given participant was only included if the participant had more than eight spectra for both 'remembered' and 'forgotten' conditions for that bin. The number of participants per time bin therefore varied (mean: n = 17.47; range: n = 12 –19).

To visualize the time course of data acquired using fMRI, for each participant, and for each time bin during the inference test trial, the time course of the preprocessed BOLD signal was extracted from the hippocampal ROI (*Figure 5A*) and from two control ROIs defined using a 12 mm sphere within our partial epi volume (*Figure 3A*). The first control region was positioned at the junction between parietal and occipital cortex ('parietal-occipital cortex') while the second control region was positioned within the brain stem (*Figure 5—figure supplement 1A, B*). For each ROI, the obtained signal for each trial was resampled using a resolution of 400 ms and regressed against an EV indicating those trials that were 'remembered'. To control for differences in baseline BOLD at the start of the trial, we also included a 'nuisance' EV indicating whether the previous trial was 'remembered'. We then plotted the normalized averaged fMRI regression coefficient for 'remembered' versus 'forgotten', using the time bins defined for the fMRS moving average (described above) (*Figure 5D*; *Figure 5—figure supplement 1A, B*).

## Acknowledgements

We would like to thank Aaron Hess for advice regarding the combined fMRI-fMRS sequence, and Saad Jbabdi and Thomas Nichols for advice on statistical approaches to analysing simulated data. RSK is supported by an EPSRC/MRC-funded studentship (EP/L016052/1). AS is supported by a Wellcome Trust studentship (203836/Z/16/Z). IBI is supported by a Royal Society Dorothy Hodgkin Research Fellowship. DD is supported by the Biotechnology and Biological Sciences Research Council UK (BBSRC UK award BB/N0059TX/1) and the MRC (Programme MC_UU_12024/3). HCB is supported by the John Fell Oxford University Press Research Fund (Grant 153/046), a seed grant from the Wellcome Centre for Integrative Neuroimaging, a Junior Research Fellowship from Merton College (University of Oxford), and the Medical Research Council (MRC) UK (MC_UU_12024/3). The Wellcome Centre for Integrative Neuroimaging is supported by core funding from the Wellcome Trust (203139/Z/16/Z).

## Additional information

### Funding

| Funder | Grant reference number | Author |
|---|---|---|
| Engineering and Physical Sciences Research Council | EP/L016052/1 | Renée S Koolschijn |
| Wellcome Trust | 203836/Z/16/Z | Anna Shpektor |
| Biotechnology and Biological Sciences Research Council | BB/N0059TX/1 | David Dupret |
| Medical Research Council | MC_UU_12024/3 | David Dupret Helen C Barron |
| John Fell Fund, University of Oxford | 153/046 | Helen C Barron |

| Funder | Grant reference number | Author |
|---|---|---|
| Wellcome Centre for Integrative Neuroimaging | Seed grant | Helen C Barron |
| Merton College, University of Oxford | JRF | Helen C Barron |
| Wellcome Trust | 203139/Z/16/Z | Renée S Koolschijn<br>Anna Shpektor<br>William T Clarke<br>I Betina Ip<br>Helen C Barron |
| Royal Society Dorothy Hodgkin Research Fellowship | | I Betina Ip |

The funders had no role in study design, data collection and interpretation, or the decision to submit the work for publication.

## Author contributions

Renée S Koolschijn, Anna Shpektor, Conceptualization, Data curation, Formal analysis, Funding acquisition, Investigation, Methodology, Software, Validation, Visualization, Writing – original draft, Writing – review and editing; William T Clarke, Formal analysis, Methodology, Software, Validation, Writing – original draft, Writing – review and editing; I Betina Ip, Formal analysis, Methodology, Writing – original draft, Writing – review and editing; David Dupret, Conceptualization, Funding acquisition, Methodology, Supervision, Writing – original draft, Writing – review and editing; Uzay E Emir, Formal analysis, Investigation, Methodology, Writing – original draft, Writing – review and editing; Helen C Barron, Conceptualization, Data curation, Formal analysis, Funding acquisition, Investigation, Methodology, Project administration, Resources, Software, Supervision, Validation, Visualization, Writing – original draft, Writing – review and editing

## Author ORCIDs

Renée S Koolschijn (iD) http://orcid.org/0000-0001-9553-4213
William T Clarke (iD) http://orcid.org/0000-0001-7159-7025
I Betina Ip (iD) http://orcid.org/0000-0003-3544-0711
David Dupret (iD) http://orcid.org/0000-0002-0040-1766
Uzay E Emir (iD) http://orcid.org/0000-0001-5376-0431
Helen C Barron (iD) http://orcid.org/0000-0002-4575-6472

## Ethics

Human subjects: All participants gave informed written consent. All experiments were approved by the University of Oxford ethics committee (reference number R43594/RE001).

## Decision letter and Author response

Decision letter https://doi.org/10.7554/eLife.70071.sa1
Author response https://doi.org/10.7554/eLife.70071.sa2

# Additional files

## Supplementary files

• Supplementary file 1. The effect of sex on behaviour and on glu/GABA ratio in primary visual cortex (V1). Using a general linear model (GLM), differences in sex (male or female) were regressed onto behavioural performance during both the inference test and associative test, and onto glu/GABA ratio during the question period of the inference test. No significant effect of sex was observed.

• Supplementary file 2. Number of trials per condition. The number of trials per condition, reported as mean ± SEM.

• Supplementary file 3. Number of trials split according to outcome. The number of inference trials per condition ('remembered' and 'forgotten', see Materials and methods for definition), split according to whether the auditory cue was indirectly associated with a rewarding or neutral outcome

(set 1: rewarded; set 2: neutral), reported as mean ± SEM. There was no significant difference in the number of trials split by set (memory set, two-way ANOVA: $F_{(1,68)}$=0.67, p = 0.424). Notably, the average difference in the total number of trials in set 1 and 2 was less than one trial, suggesting memory recall was not confounded by reward status.

• Supplementary file 4. Functional magnetic resonance imaging (fMRI) contrast for 'remembered'–'forgotten'. The fMRI blood oxygen level-dependent (BOLD) signal was assessed for a contrast comparing 'remembered' and 'forgotten' trials (*Figure 3B*) in the inference test. Brain regions that survived whole-volume correction for multiple comparisons are listed (p < 0.05 with whole-brain family wise error [FWE] correction at the cluster level). Montreal Neurological Institute (MNI) coordinates are listed. Notably, the activation in hippocampus was bilateral but the peak on the right-hand side was not significant when applying whole-brain FWE correction ($t_{17}$ = 3.74, p > 0.05 [x = 22, y=-24, z=−13]).

• Supplementary file 5. Inter-subject covariance of glutamate and GABA. Inter-subject covariances (%) for the key metabolite measurements during the 'Question' period of inference trials (presented in *Figure 4D–E*).

• Supplementary file 6. Average number of spectra (NEX). The number of spectra contributing to metabolite estimates during the various trial periods in the inference test (mean ± SEM).

• Supplementary file 7. Covariance between hippocampal blood oxygen level-dependent (BOLD) signal and functional magnetic resonance spectroscopy (fMRS) for remembered versus forgotten. The relationship between functional magnetic resonance imaging (fMRI) and fMRS during 'remembered' versus 'forgotten' trials in the inference test was assessed. To this end, fMRS measures of glu/GABA ratio from primary visual cortex (V1) for 'remembered'–'forgotten' were included as covariates in a group analysis for the equivalent fMRI contrast (p < 0.05 with family wise error [FWE] correction at the cluster level). The only brain region to survive whole-brain correction for multiple comparisons was the left hippocampus. Thus, the BOLD signal in left hippocampus significantly predicted individual differences in glu/GABA ratio measured from V1 during memory recall; MNI coordinates.

• Supplementary file 8. Minimum reporting standards in MRS (MRSinMRS) checklist.

• Transparent reporting form

### Data availability

All data generated and analysed during this study are included in the manuscript and supporting files. The data and code used in this study are available via the MRC BNDU Data Sharing Platform upon registration. The data is available via https://data.mrc.ox.ac.uk/data-set/fmri-fmrs-inference. The code is available via https://data.mrc.ox.ac.uk/data-set/frms-code.

The following dataset was generated:

| Author(s) | Year | Dataset title | Dataset URL | Database and Identifier |
|---|---|---|---|---|
| Koolschijn RS, Shpektor A, Clarke WT, Betina Ip I, Dupret D, Emir UE, Barron HC | 2021 | Combined fMRI-fMRS dataset in an inference task in humans | https://doi.org/10.5287/bodleian:vmJOOm7KD | MRC BNDU Data sharing platform, 10.5287/bodleian:vmJOOm7KD |
| Koolschijn RS, Shpektor A, Clarke WT, Dupret D, Emir UE, Barron HC | 2021 | Code used for analysis of task-relevant, time-resolved functional Magnetic Resonance Spectroscopy (fMRS) | https://doi.org/10.5287/bodleian:8JwYayQmD | MRC BNDU Data Sharing Platform, 10.5287/bodleian:8JwYayQmD |

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

# Appendix 1

## Supplementary note 1

For the fMRS quantification, we implemented an analysis pipeline in LCModel that is optimised for detecting dynamic changes in GABA. Consequently, we do not implement default assumptions typically used to obtain static estimates of GABA, where metabolite values are subject to prior constraints (or 'soft constraints') that ensure values are within a predefined ('physiologically plausible') range.

Default LCModel settings impose prior constraints using the following concentration ratio:

$$\frac{[\text{GABA}]}{[\text{Big3}]} = 0.04 \pm 0.04 ,$$

$$[\text{Big3}] = [\text{totNAA}] + [\text{totCr}] + 3[\text{totCho}] .$$

Implemented as 'CHRATO(9)' in LCModel, where a Gaussian prior is imposed on the concentration of GABA with a mean and standard deviation of 0.04 times that of a weighted sum of total *N*-acetylaspartate (totNAA), total creatine (totCr), and total choline (totCho) concentrations. These default priors necessarily bias the estimated concentration of GABA, whilst reducing variance. These default priors are therefore not appropriate when detecting dynamic changes in GABA.

