## [Decision Letter]

**Acceptance summary:**

This is a novel and timely investigation of the balance between excitation and inhibition to explore the role of glutamate and GABA during memory retrieval. The innovative use of rapidly interleaved fMRI and fMRS provides a compelling link between successful retrieval effects in hippocampus and inhibitory/excitatory dynamics in visual cortex. The study itself is well-motivated and well executed, complementing prior cross-species work, and provides an intriguing set of results to support the major claims.

**Decision letter after peer review:**

Thank you for submitting your article "Memory recall involves a transient break in excitatory-inhibitory balance" for consideration by *eLife*. Your article has been reviewed by 2 peer reviewers, and the evaluation has been overseen by a Reviewing Editor and Chris Baker as the Senior Editor. The following individuals involved in review of your submission have agreed to reveal their identity: Paul Mullins (Reviewer #1); Ben Hutchinson (Reviewer #2).

Essential revisions:

1. Please explain the timeline of MRS data acquisition throughout the experimental paradigm a little better. I assume that data acquisition and experimental paradigm were started at the same time, but given the MRS data were collected in step sizes of 4 secs, it may be hard to understand how a temporal resolution of 2.5 secs for the fMRS data is achieved. The experimental blocks varied in their onset and duration leading to the MRS data being acquired at different points within the experimental blocks.

2. Likewise, given that the total Question and ITI time was allowed to vary from stimulus to stimulus, it is difficult to understand how the timeline in figures 4 and 5 are achieved. This could be ameliorated by a better explanation of data collection process in the methods, and how data was averaged to produce the timelines as presented.

3. Please include the checklist from the recent Minimum Reporting Standards in Magnetic Resonance Spectroscopy (MRSinMRS) in the supplementary material (Citation included below). While the methods present most of the pertinent pieces of information, the inclusion of the completed checklist will ensure all relevant information is included, provide a concise record for readers to follow exactly how the MRS data was collected, as well as follow current standards for reporting:

Lin, Alexander, Ovidiu Andronesi, Wolfgang Bogner, In-Young Choi, Eduardo Coello, Cristina Cudalbu, Christoph Juchem, et al., 'Minimum Reporting Standards for in vivo Magnetic Resonance Spectroscopy (MRSinMRS): Experts' Consensus Recommendations'. NMR in Biomedicine n/a, no. n/a (n.d.): e4484. https://doi.org/10.1002/nbm.4484.

4. A description of what criteria were used for assessing data quality would be useful (particularly in the case where a participant was excluded).

5. It was difficult to understand the nature of the videos accompanying the tone probes during the inference test. Were these just random segments of videos of the virtual environment or were they of the background for the relevant visual cue?

6. The robustness of the results might be informed by some sort of assessment of reliability based on the multiband EPI sequence which was also collected (on the same task?) and described in another paper.

7. Does the ratio of rewarded versus neutral trials remain roughly even across remembered and forgotten conditions when conditionalized on both the inference and associative memory task (i.e. the main way the data was carved up for the analyses)? The concern would be if there is a confound between memory success and reward status, then further analyses might be warranted.

8. Although the voxelwise 'reverse' correlation between the fMRS results and the BOLD data were compelling, it would be good to also show the (presumably lack of) correlation between the BOLD and MRS results for the two control ROIs in Figure S10.

9. Is there any reason why the data for glutamate and GABA is plotted on the same y axis or different y axes across panels in Figure 4?

10. What coordinate system (e.g. MNI) is being used for reporting the BOLD results?

11. On line 300 it is stated that "the hippocampal BOLD signal positively predicted the relative concentration of glutamate (trend)," but then the statistics reported in the caption for Figure 5b suggest the p value of the relationship is.585.

12. At times, the number of observations used in the analyses is unclear. For example, in the caption of Figure 5, there are 16 degrees of freedom for the correlations (implying N = 18), 16 degrees of freedom for the voxelwise t-test (implying N = 17 for a paired test, although perhaps a different test was used), and a description of the moving average which states N = 19.

---

## [Author Response]

Essential revisions:1. Please explain the timeline of MRS data acquisition throughout the experimental paradigm a little better. I assume that data acquisition and experimental paradigm were started at the same time, but given the MRS data were collected in step sizes of 4 secs, it may be hard to understand how a temporal resolution of 2.5 secs for the fMRS data is achieved. The experimental blocks varied in their onset and duration leading to the MRS data being acquired at different points within the experimental blocks.

As the reviewers note, the data acquisition and experimental paradigm were started at the same time. A temporal jitter was included in each trial of the experimental paradigm, by varying the length of the auditory tone (mean of 7 s, minimum of 4 s and maximum of 14 s, sampled from a truncated γ distribution), the participant’s response (self-paced) and the inter-trial interval (mean of 2.7 s, minimum of 1.4 s, maximum of 10 s, sampled from a truncated γ distribution). Due to this jitter the temporal relationship between the data acquisition and the experimental paradigm varied across trials, i.e. the fMRS was acquired at a different time point in every trial. Thus, over the duration of the scan, spectra were acquired in each segment of the inference test trial and it was therefore possible to generate a moving average for the fMRS at a higher temporal resolution than the TR of 4.105 s. This is now made clear in the *Results* and *Methods* sections, together with a schematic in the new Figure 4—figure supplement 1 which is referenced in the legend to Figures 4, 5 and Figure 5—figure supplement 1.

2. Likewise, given that the total Question and ITI time was allowed to vary from stimulus to stimulus, it is difficult to understand how the timeline in figures 4 and 5 are achieved. This could be ameliorated by a better explanation of data collection process in the methods, and how data was averaged to produce the timelines as presented.

See our response to point #1 above. To clearly explain this point we have edited the *Results* and *Methods* sections and include a schematic illustrating the temporal relationship between data acquisition and the experimental paradigm in the new Figure 4—figure supplement 1.

3. Please include the checklist from the recent Minimum Reporting Standards in Magnetic Resonance Spectroscopy (MRSinMRS) in the supplementary material (Citation included below). While the methods present most of the pertinent pieces of information, the inclusion of the completed checklist will ensure all relevant information is included, provide a concise record for readers to follow exactly how the MRS data was collected, as well as follow current standards for reporting:Lin, Alexander, Ovidiu Andronesi, Wolfgang Bogner, In-Young Choi, Eduardo Coello, Cristina Cudalbu, Christoph Juchem, et al., 'Minimum Reporting Standards for in vivo Magnetic Resonance Spectroscopy (MRSinMRS): Experts' Consensus Recommendations'. NMR in Biomedicine n/a, no. n/a (n.d.): e4484. https://doi.org/10.1002/nbm.4484.

Thank you for this important point. We have now included this checklist in the new Supplementary File 8.

4. A description of what criteria were used for assessing data quality would be useful (particularly in the case where a participant was excluded).

Thanks for this point. We have summarised how data quality were assessed below:

MRS data quality: The quality of MRS data was assessed during set-up scans and acquisition. Using this approach, several criteria were considered: we monitored the level of noise in the data and the residual water signal to check for stability and to ensure that the size of the water peak was well below the level of most observable metabolites. During set-up scans, the position of the MRS voxel was adjusted if necessary.

Data quality was further assessed during data analysis, using the metrics reported in Figure 4—figure supplement 3 and listed in Supplementary File 5 (i.e. tCr linewidth, FWHM, SNR, CRLB). No participants were excluded due to poor MRS data quality.

fMRI data quality: As noted in the *Methods* section, the quality of the fMRI data in the combined fMRI-fMRS sequence was compromised relative to non-combined state-of-the-art EPI sequences. Across participants, the quality of the data was assessed during the pre-processing of the data. Namely, if the quality of the data was insufficient to allow co-registration between the EPI and structural scan, then data were excluded. This applied to data from one participant, as stated in the *Methods* section.

In the *Methods* sections “*fMRS data acquisition*” and “*fMRI preprocessing and GLMs*” we have added a brief summary of the above two paragraphs.

5. It was difficult to understand the nature of the videos accompanying the tone probes during the inference test. Were these just random segments of videos of the virtual environment or were they of the background for the relevant visual cue?

Each video during the inference test consisted of a pseudo-random trajectory through the virtual environment. The videos were not related to the background for the relevant visual cue. This is now clarified in the *Methods* section. Moreover, as noted in the *Methods* section, to control for potential confounding effects of space, each video involved a trajectory constrained to a 1/16 quadrant of the VR environment, evenly distributed across the different auditory cues. To assign the videos to each trial in the inference test randomisation was performed separately for each participant to ensure no consistent biases. An example video is now included in the supplementary materials (Figure 1–Video 1).

6. The robustness of the results might be informed by some sort of assessment of reliability based on the multiband EPI sequence which was also collected (on the same task?) and described in another paper.

Thanks for this suggestion. The reported fMRI main effect (increase in BOLD for correctly inferred versus incorrectly inferred trials) in visual cortex and hippocampus (Figure 3C) was indeed observed in equivalent analyses conducted on data acquired on the same task using a multiband EPI sequence (Barron et al., Cell 2020). Moreover, the higher quality multiband data acquired previously permitted a searchlight Representational Similarity Analysis (RSA) which further revealed reinstatement of the associated visual cues in hippocampus and visual cortex during inference. This latter result further demonstrates the involvement of these two brain regions in memory recall and provided the basis for investigating the underlying mechanism for memory recall using the data reported here. In the revised *Results section* we now refer to these previous results in more detail. We also include a new paragraph in the *Discussion section* that further highlights how the replication across data sets and consistency with other research indicates the reliability of the reported findings.

7. Does the ratio of rewarded versus neutral trials remain roughly even across remembered and forgotten conditions when conditionalized on both the inference and associative memory task (i.e. the main way the data was carved up for the analyses)? The concern would be if there is a confound between memory success and reward status, then further analyses might be warranted.

Thanks for this interesting point. We did not observe a significant difference in the number of set 1 (rewarded) and set 2 (neutral) trials across ‘remembered’ and ‘forgotten’ conditions when conditionalized on both the inference and associative memory task (memory x set, two-way ANOVA: F_(1,68)_=0.67, p=0.424). This ANOVA together with the raw data (mean number of trials ± SEM) is now reported in the new Supplementary File 3 which is referenced in the revised *Results section*.

8. Although the voxelwise 'reverse' correlation between the fMRS results and the BOLD data were compelling, it would be good to also show the (presumably lack of) correlation between the BOLD and MRS results for the two control ROIs in Figure 5–figure supplement 1.

Thanks for this suggestion. Across participants the difference in BOLD signal in brainstem and parietal cortex during ‘remembered’ compared to ‘forgotten’ trials did not significantly predict the increase in glu/GABA ratio observed in V1. These results are now shown in the revised Figure 5—figure supplement 1 and the corresponding correlation coefficients reported in the accompanying figure legend.

9. Is there any reason why the data for glutamate and GABA is plotted on the same y axis or different y axes across panels in Figure 4?

We apologize if the different y axes caused any confusion. The combined/split y-axis for Figure 4A vs Figure 4C is for visualisation purposes only. We would be happy to split the y-axis in Figure 4A or combine the y-axis in Figure 4C if the reviewers prefer. However, we believe the current lay-out is the most suitable for illustrating the dynamics of glutamate and GABA in a clear manner.

10. What coordinate system (e.g. MNI) is being used for reporting the BOLD results?

Apologies for this oversight. We used the MNI coordinate system throughout. This is now stated in the legend to Figures 3 and 5; Figure 3—figure supplement 1 and Figure 5—figure supplement 1; Supplementary Files 4 and 7.

11. On line 300 it is stated that "the hippocampal BOLD signal positively predicted the relative concentration of glutamate (trend)," but then the statistics reported in the caption for Figure 5b suggest the p value of the relationship is .585.

Apologies for this mistake. We have now updated this section of the *Results* accordingly, which now reads:

“In line with this prediction, across participants the hippocampal BOLD signal negatively predicted the relative concentration of GABA, and positively predicted the increase in glu/GABA ratio in V1 (‘remembered’ versus ‘forgotten’ trials; Figure 5A-B).”

12. At times, the number of observations used in the analyses is unclear. For example, in the caption of Figure 5, there are 16 degrees of freedom for the correlations (implying N = 18), 16 degrees of freedom for the voxelwise t-test (implying N = 17 for a paired test, although perhaps a different test was used), and a description of the moving average which states N = 19.

We would like to thank the reviewers for highlighting this point of confusion. The participant excluded from the fMRS analysis (n=1 in the ‘Question’ period, due to there being less than 8 spectra for either ‘remembered’ or ‘forgotten’) was not the same participant as the one excluded from the fMRI analysis (n=1, where data quality was not sufficient to ensure reliable pre-processing). This is now stated in the *Methods*. Analyses that assess the relationship between fMRS and fMRI data therefore include n=17 participants.

Thus, we note that there were indeed 16 degrees of freedom for the voxelwise t-test reported in Figure 5C. However, for the correlations reported in Figure 5B the degrees of freedom should have been reported as 15. This has now been corrected.

We also note that for the moving average used to visualize the data in Figure 4C and 5D, the degrees of freedom varied by time bin, with a maximum of n=19 participants per bin. In response to the reviewers’ points #1 and #2, we have revised the section of the *Methods* that explains how we estimated the moving average. In addition, we now state the mean number of participants included in each time bin.

We hope the reviewers agree that the number of observations used in each analysis is now clear.